# MSBooster: improving peptide identification rates using deep learning-based features

Kevin L. Yang [1], Fengchao Yu [2] ✉, Guo Ci Teo [2], Kai Li[1], Vadim Demichev[3,4], Markus Ralser [3,5,6] & Alexey I. Nesvizhskii [1,2] ✉

Peptide identification in liquid chromatography-tandem mass spectrometry (LC-MS/MS) experiments relies on computational algorithms for matching acquired MS/MS spectra against sequences of candidate peptides using database search tools, such as MSFragger. Here, we present a new tool, MSBooster, for rescoring peptide-to-spectrum matches using additional features incorporating deep learning-based predictions of peptide properties, such as LC retention time, ion mobility, and MS/MS spectra. We demonstrate the utility of MSBooster, in tandem with MSFragger and Percolator, in several different workflows, including nonspecific searches (immunopeptidomics), direct identification of peptides from data independent acquisition data, single-cell proteomics, and data generated on an ion mobility separation-enabled timsTOF MS platform. MSBooster is fast, robust, and fully integrated into the widely used FragPipe computational platform.

Liquid chromatography–tandem mass spectrometry (LC−MS/MS) is an established, widely used high-throughput method for elucidating the proteome[1]. In the typical LC−MS/MS proteomic workflow, proteins are extracted from the samples and digested into peptides, most commonly using trypsin, which cleaves after lysine and arginine residues. For complex samples, if a high depth of protein identification is required, the workflows are combined with fractionation or enrichment techniques (e.g., to increase the detection of phosphorylated peptides). The peptide preparations are then separated using LC coupled online to a mass spectrometer, and the peptides eluting from the LC column are ionized and transferred to the gas phase. The mass-to-charge (m/z) values of all peptide ions from all peptides eluting from the LC column at a particular retention time (RT) are measured using the first stage of MS, generating an MS1 spectrum. These spectra contain the m/z values of all detectable ions and their intensities. Optionally, ions can also be separated using ion mobility (IM). In the second stage of MS analysis, selected (typically the most intense) peptide ions are subjected to isolation and fragmentation to break the peptide bonds; this approach is called data-dependent acquisition (DDA)[2]. Alternatively, all peptide ions within a wider window of m/z

values or in a continuous quadrupole scan of a particular window size[3] are selected for simultaneous fragmentation; this approach is called data-independent acquisition (DIA)[4]. The resulting MS/MS or MS2 spectra, whether generated in the DIA or DDA mode, contain m/z values, intensities, and sometimes IM values of all observed fragment ions for the precursor peptides subjected to MS/MS.

The acquired MS/MS spectra, along with their RT, IM, and corresponding precursor peptide masses for DDA or mass windows for DIA, are used to identify the sequences of the peptides that generated the spectra[5]. This is typically done using the sequence database search approach. Computational tools such as MSFragger[6,7], SEQUEST[8], Andromeda[9], MASCOT[10], MetaMorpheus[11], and Comet[12] compare each experimental MS/MS spectrum against a set of theoretical m/z values of fragments calculated for each candidate peptide based on the provided protein sequence database and assign a score for each peptide-to-spectrum match (PSM). Not every top-scoring PSM is a correct identification. These mismatches may be a result of noise in the spectra or true peptide sequences missing in the provided protein sequence database[5,13,14]. To assist with downstream false discovery rate (FDR) control, decoys are typically added, where decoys are shuffled or

[1]Department of Computational Medicine and Bioinformatics, University of Michigan, Ann Arbor, MI, USA. [2]Department of Pathology, University of Michigan, Ann Arbor, MI, USA. [3]Department of Biochemistry, Charité Universitätsmedizin, Berlin, Germany. [4]Department of Biochemistry, University of Cambridge, Cambridge, UK. [5]Nuffield Department of Medicine, The Wellcome Centre for Human Genetics, University of Oxford, Oxford, UK. [6]Max Planck Institute for Molecular Genetics, Berlin, Germany. ✉e-mail: yufe@umich.edu; nesvi@med.umich.edu

reversed versions of sequences from the "target" protein database[5,15,16]. Search engines output a list of PSMs, which are used as input to computational post-processing tools such as PeptideProphet[5,17,18] and Percolator[19,20], which combine various search engine scores, such as the hyperscore and expectation value, and other properties that are useful for discrimination, such as the difference between the theoretical mass of the peptide and the mass derived from the measured *m/z*. The differences in the distributions of scores for decoy peptides versus those of target peptides are used as part of the modeling process to determine the optimal combination of individual features, as well as to calculate posterior probabilities of correct identification and estimate FDR. These tools significantly boost the sensitivity of peptide and protein identification at a fixed FDR compared with filtering the data using individual scores reported by the search engine[5].

Although tools such as PeptideProphet and Percolator are now a part of many computational pipelines, including FragPipe, they do not incorporate prior knowledge regarding peptide separation coordinates (RT, IM) or fragment ion intensities. High-confidence PSMs from previously published studies are stored in public repositories and can be leveraged via spectral library searching[21–26], in which known fragment ion intensities help differentiate true from false PSMs. However, relying on experimentally derived spectral libraries is often limiting, as these libraries are inherently incomplete. For instance, protein expression varies from biological condition to condition, cell type to cell type, and genetic background to background, so libraries can be incomplete even for organisms with large amounts of previous MS/MS data available. Thus, approaches for predicting MS/MS spectra[27] and using predicted spectra from available protein sequence data to improve the sensitivity of peptide identification in LC−MS/MS proteomics have been explored[28–30]. The difference between the experimental and predicted retention times is also known to provide additional discriminating power[31–34]; RT differences have previously been incorporated into Percolator and PeptideProphet modeling[34,35]. However, the use of RT and MS/MS spectral predictions was initially limited, in part because of the limitations of first-generation prediction algorithms.

More recently, however, a wave of deep learning (DL) models has been trained to predict the physicochemical properties of peptides and MS/MS spectra[36–41]. By training on millions of available peptides, these models can learn general rules to make accurate predictions for new peptides, assuming they are not vastly different from those on which the models were trained. The use of DL-based RT and spectral predictions have been shown to be particularly useful for DIA data analysis[42–44], and for improving the identification rates in immunopeptidome studies concerned with the analysis of human leukocyte antigen (HLA) binding peptides[45–48]. Unfortunately, current PSM rescoring tools that take advantage of DL-based predictions may be difficult for some users to adopt. For example, MaxQuant with Prosit rescoring requires users to upload their database search results to a web server. Rescoring may be performed locally if the users have GPU access, which is not always the case. DeepRescore[49] requires Docker and Nextflow, which may be difficult for users with less computational experience to install.

Here, we present the DL-based PSM rescoring tool MSBooster, a new addition to the widely used FragPipe computational platform. MSBooster provides a fully automated and integrated solution for the use of DL predictions for improved peptide and protein identification. It uses a DL model to predict the RT, IM, and MS/MS spectra of peptides, followed by the generation of additional features for PSM rescoring with Percolator[19]. No external prediction of spectral libraries is required, bypassing concerns about uploading data to shared servers and data privacy. We demonstrate the flexibility of MSBooster and its performance in several different workflows, including HLA immunopeptidome nonspecific searches, DIA quantitative proteomics, single-cell proteomics, and data generated on an IM-enabled timsTOF

MS platform. We also explored the behavior of spectral and RT features in the analysis of single-cell proteomics data and investigated the potential benefits of using multiple correlated similarity metrics in Percolator. Finally, we assess and discuss the utility of incorporating IM predictions into PSM rescoring.

## Results

### MSBooster and FragPipe computational workflow

FragPipe (https://fragpipe.nesvilab.org/) is a comprehensive computational platform that automates all steps of proteomic analysis, including peptide-spectrum matching with MSFragger[6,50,51], PSM validation with PeptideProphet[18] or Percolator[19], protein inference with ProteinProphet[52], and FDR filtering (by default 1% FDR at the PSM, ion, peptide, and protein levels) using Philosopher[53]. FragPipe supports the generation of spectral libraries using EasyPQP (https://github.com/grosenberger/easypqp) and the extraction of quantification from DIA data (using DIA-NN[42,43]). DIA-Umpire[54] is included in FragPipe as one of the modules to generate pseudo-MS/MS spectra from the DIA data. Alternatively, peptides can be identified directly from DIA data using MSFragger-DIA[55]. FragPipe has an easy-to-use graphical user interface (GUI) and includes a data visualization module (FragPipe-PDV), which is an extension of a previously described PDV viewer[56].

Within FragPipe, MSBooster is positioned between MSFragger and Percolator (Fig. 1a) and is enabled by default in most FragPipe analysis workflows (see the "Methods" section for details), where a FragPipe workflow is the order in which software is to be executed, along with optimized parameters for each tool. MSBooster's role can be divided into the separate steps of peptide extraction from PSM files, input file formatting for a DL model, feature calculation using observed and predicted peptide properties, and addition of the new features to the PSM files (Fig. 1b). In a typical workflow, MSFragger performs the database search and reports the list of PSMs and associated search scores in a "pin" file. Without MSBooster, these pin files are passed directly to Percolator. When MSBooster is enabled, it extracts the set of peptides reported in the pin file and creates an input file for a DL model, which in turn generates predictions of the physicochemical properties of peptides, namely RT, IM, and/or MS/MS spectra. Within FragPipe, we chose DIA-NN[42,43] to predict these properties, as it is already included for DIA quantification. We also show the compatibility of Prosit predictions with standalone MSBooster, although this is not yet supported in FragPipe. Importantly, predictions are performed only for the relatively small set of PSM candidate peptides reported by MSFragger, rather than the whole in-silico digested proteome. Thus, predictions can be done for each dataset on-the-fly, without the need for time-consuming full spectral library prediction. MSBooster then generates features based on the agreement between the experimental and predicted values and adds them to the original pin files. Finally, it passes these extended pin files to Percolator, which learns a linear support vector machine (SVM) to differentiate true targets from decoys[19]. Percolator assigns an SVM score and then a posterior error probability to each PSM. DL-based predictions are done for a limited number of peptide candidates: by default, either for a single (top scoring) peptide per MS/MS spectrum (DDA data), top 3 (narrow window GFP-DIA data), or top 5 (conventional DIA data) when using MSFragger-DIA. Thus, in most cases, MSBooster resulted in only a minor increase in the overall computational run time (Supplementary Fig. 1).

### HLA peptide identification

Immunopeptidomics, that is, methods that identify and quantify peptides that are presented as antigens by antigen-presenting cells, are increasingly required in biomedicine but are associated with computational challenges. Human leukocyte antigen (HLA) peptidome data is a promising candidate for DL-based rescoring owing to an expanded nonenzymatic search space, resulting in a higher probability of a high-

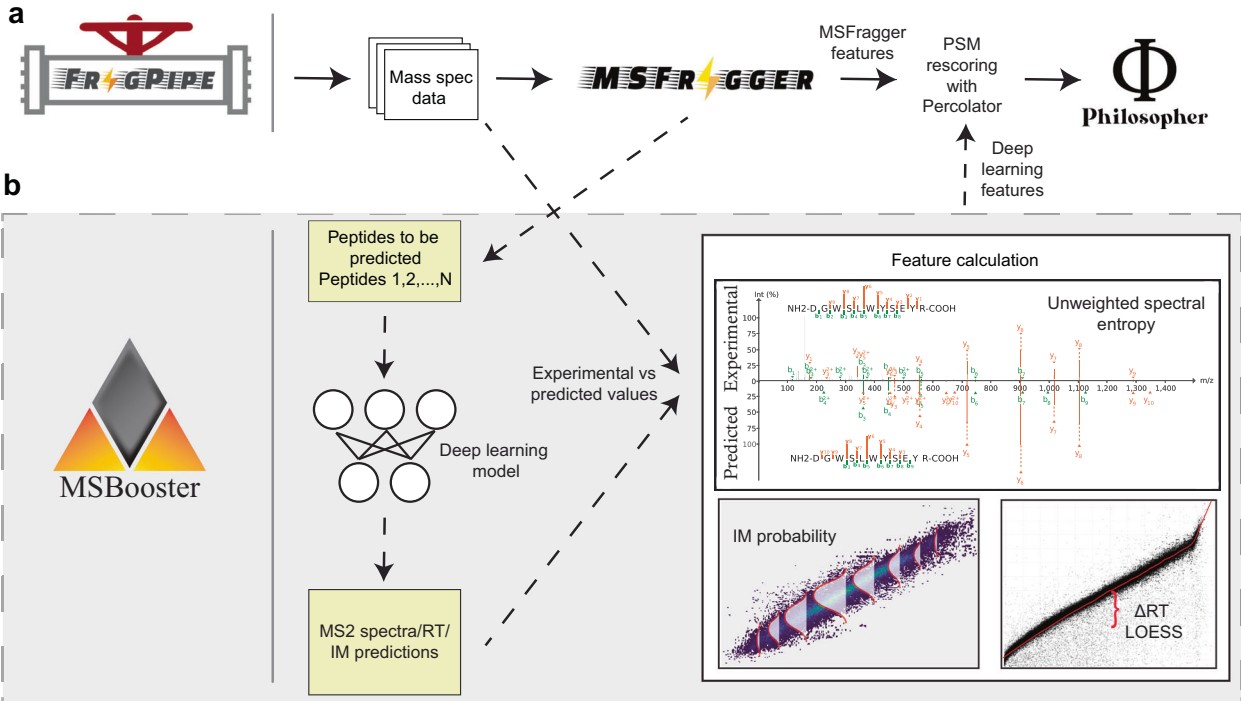

**Fig. 1 | MSBooster workflow.** The original workflow without MSBooster (**a**) and the new workflow with MSBooster (**b**) are depicted. Files generated when running MSBooster are depicted in yellow. Dashed arrows are steps run only when using MSBooster. The default features used by MSBooster are shown in "Feature calculation". Features reported by MSFragger, such as hyperscore and charge, are combined with deep learning features calculated in MSBooster for Percolator rescoring before filtering and reporting in Philosopher.

scoring false match. Because certain major histocompatibility complexes (MHCs) preferentially bind certain peptide motifs, this represents a system in which we know what kinds of peptides should be identified based on their sequences, allowing us to monitor whether MSBooster correctly promotes true target PSMs. To demonstrate the performance of MSBooster with MS/MS spectral and RT-based rescoring on HLA peptides, three fractions of an A*02:01 monoallelic cell line[57] were processed using different combinations of features in MSBooster. MS/MS data were acquired on an Orbitrap Exploris 480 (Thermo Fisher Scientific) with higher energy collisional dissociation (HCD). Spectral and RT features increased the number of identified peptides by 20.4% and 16.6%, respectively, at 1% FDR, whereas the combination of the two feature types increased the number of identifications by 31.4% (Fig. 2a). Each addition of a new DL feature resulted in a statistically significant increase in peptides (*t*-statistics and *p*-values for each comparison are noted in Supplementary Data 1).

HLA peptide rescoring with DL features has previously been explored by Wilhelm et al. [46]. The authors showed an average increase of 159% in peptide identification across 92 monoallelic cell lines[58] when using MaxQuant coupled with Prosit rescoring (Supplementary Fig. 3a). While this may make the 31.4% increase with MSBooster on the Klaeger et al. data seem minimal, this discrepancy may simply reflect the moderate performance of MaxQuant in nonspecific searches, as noted by Parker et al. [59], and its use of only the Andromeda score for ranking PSMs before FDR filtering[9]. Even without DL-based rescoring, MSFragger provides multiple discriminative scores (Supplementary Fig. 2a). To provide a more accurate comparison, we rescored PSMs using only the hyperscore––MSFragger's database search score–as a starting point (Supplementary Fig. 3b). Giving Percolator access to only the hyperscore, we reported an average of 2949 peptides. Adding spectral and RT features to the hyperscore provided a 183.8% increase in the number of identified peptides, a statistic more in line with the 159% increase reported by Wilhelm et al.[46]. Importantly,

adding other features reported by MSFragger (Supplementary Fig. 2a) also gives a 161.5% boost compared to using MSFragger's hyperscore alone, indicating the utility of non-DL features. Using all MSFragger computed and all DL-based from MSBooster features together results in a 243.7% boost compared with using the hyperscore alone.

We then specifically compared the performance between MSFragger/MSBooster and MaxQuant/Prosit on the Klaeger et al. data. MaxQuant initially reported 1569 peptides. After PSM rescoring with Prosit, it reported 10,680 peptides, a 681% increase. This is compared to the 10,138 peptides reported with MSBooster rescoring averaged across 10 Percolator runs (Supplementary Fig. 3b). To see if MaxQuant's performance could be attributed to Prosit's predictions, we configured MSBooster to accept Prosit-predicted spectral libraries (see the "Methods" section). Indeed, using Prosit in lieu of DIA-NN allowed MSBooster to achieve 10,798 peptides on average, 118 more than MaxQuant with Prosit (Supplementary Fig. 3b).

While most reported peptides passing 1% peptide-level FDR were shared regardless of whether MSBooster was used (7016 peptides), adding MSBooster resulted in 2656 more identified peptides while only losing 262 (Fig. 2b). To verify that the added peptides were credible, we identified their HLA sequence motifs using GibbsCluster[60] (Fig. 2c and Supplementary Fig. 3c). MHC binding in the A*02:01 cell line relies on anchors at position 2 and the C-terminus, according to the Immune Epitope Database[61]. The 262 peptides lost after rescoring, when used as input in the motif analysis tool, produced two clusters in GibbsCluster. The first cluster of 132 peptides followed the expected motif, but the second cluster of 60 peptides was not enriched for the expected amino acids at position 2 (Supplementary Fig. 3c). Therefore, many of the peptides removed with the help of MSBooster were likely false positives. In contrast, peptides gained with MSBooster generated one cluster of 2533 peptides that faithfully followed the expected sequence motif for the cell line (Fig. 2c). To further validate the new peptides, we examined their binding affinities with A*02:01 MHC using

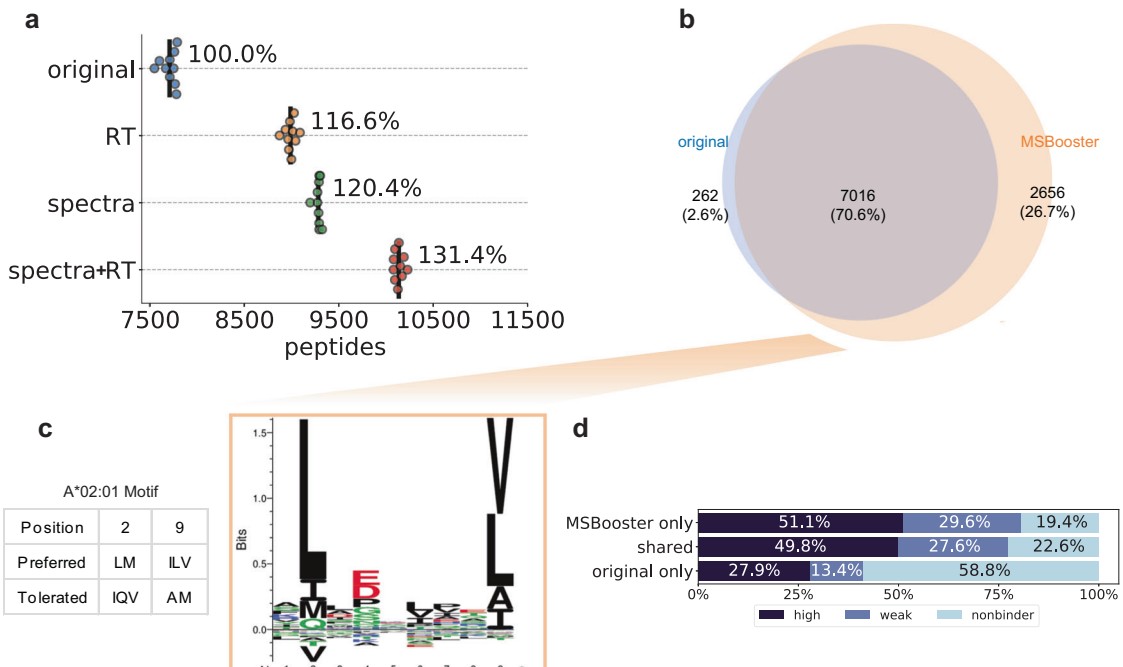

**Fig. 2 | HLA rescoring. a** Swarmplot of the number of HLA peptides reported at 1% FDR using the MSFragger pin files (original), files with the spectral similarity feature added (spectra), retention time similarity feature (RT), or both types of features (spectra+RT). Each dot represents the number reported for each of the 10 Percolator runs. Black lines show the average number of peptides reported across 10 Percolator runs. **b** Venn diagram of HLA peptides between lengths 7 and 12 when using either original MSFragger features or additional deep learning features. **c** GibbsCluster-generated motif assigned to the MSBooster-specific peptide subset from (**b**). The A*02:01 motif was collected from the Immune Epitope Database. **d** Percent of peptides from each subset of **b** that are predicted by NetMHC 4.0 to bind the A*02:01 serotype. Strength of the ligand binding decreases from "high" to "weak" to "nonbinder". Source data are provided as a Source Data file.

predictions from NetMHC[62] (Fig. 2d). We found that 2143 (80.7%) of the gained peptides and 5430 (77.4%) of the shared peptides were predicted as either strong or weak binders by NetMHC, while this percentage dropped to 41.3% (108 peptides) for the MSBooster-removed original peptides. This further supports the idea that peptides gained with DL-based rescoring in MSBooster are more reliable than those that are removed.

An important feature of MSBooster is its ability to handle peptides with post-translational modifications (PTMs) that are not predicted by the DL spectral prediction model. In DIA-NN v1.8, cysteine carbamidomethylation, methionine oxidation, N-terminal acetylation, phosphorylation, and ubiquitination are supported. A multitude of other biologically relevant PTMs exist; for example, cysteinylation is an important PTM to consider in immunopeptidomics, as it plays a role in T cell recognition[63]. Rather than precluding the inclusion of other PTMs in the search or rescoring steps, MSBooster obtains the predicted spectrum for the unmodified peptide and shifts the $m/z$ values of the PTM-containing fragments while retaining their predicted intensities. RT values are the same as those of their peptide counterparts, excluding the new PTMs (e.g., a jointly biotinylated and phosphorylated peptide will use the RT of the phosphorylated peptide). To explore how fragment peak shifting affects the results, we examined the distributions of spectral and RT feature scores for accepted PSMs after Philosopher filtering (Supplementary Fig. 4a-b). Each group of PSMs contained only the PTM listed (i.e., the PSMs in the carbamidomethylated C group were matched to peptides that only contained that PTM, and no oxidized M). As expected, unmodified, carbamidomethylated C, and oxidized M PSMs had high spectral similarities and low RT differences, since DIA-NN included them in the training set. Interestingly, although acetylated N-terminal peptides were in the training set, their spectral similarity score distributions were lower than those of the other peptides. Cysteinylation had a similar

distribution of PTMs on which DIA-NN was trained. This could mean that cysteinylation does not have a major impact on fragment intensity or that cysteinylated PSMs with lower scores were excluded after FDR filtering. PSMs with pyro-glutamation events from Q had the worst distribution of the PTMs considered. The RT shift in pyro-Glu peptides is expected and has been previously recapitulated[64]. While most PTMs showed an increase in peptides containing those modifications (Supplementary Fig. 4c), the exception is pyro-glutamation from Q, where reported peptides dropped from 105 to 25. This decrease is mainly driven by the increased RT difference, as no unique peptides are reported after only rescoring with the RT feature while rescoring with only spectral similarity results in only 32 lost peptides and 9 gained peptides. This is in stark contrast to the 82 lost and 2 gained peptides after using the MSBooster default of spectral and RT feature rescoring. Overall, our analysis shows that while MSBooster will not exclude any PTM-containing peptide, those PTMs not yet supported by the DL prediction module that drastically affect the peptides' physicochemical properties will be heavily penalized.

## Neoantigen discovery

The discovery of patient-specific neoantigens, such as those derived from genomic alterations in cancer cells, potentially represents a step up in difficulty in data analysis. For example, such clinical samples can include up to six MHC alleles instead of the one MHC allele from the monoallelic cell line in[57]. We tested MSBooster on a tissue sample from a patient with metastatic malignant melanoma (Mel15)[65]. This data was acquired on a Q Exactive instrument (Thermo Fisher Scientific, Bremen) with HCD fragmentation. Considering just canonical peptides from the UniProt database, MSBooster increased peptide identifications from 34,648 to 41,236, a 19.0% boost (Fig. 3a, Supplementary Data 2). Importantly, when considering just noncanonical peptides with variants derived from exome sequencing[65], peptide

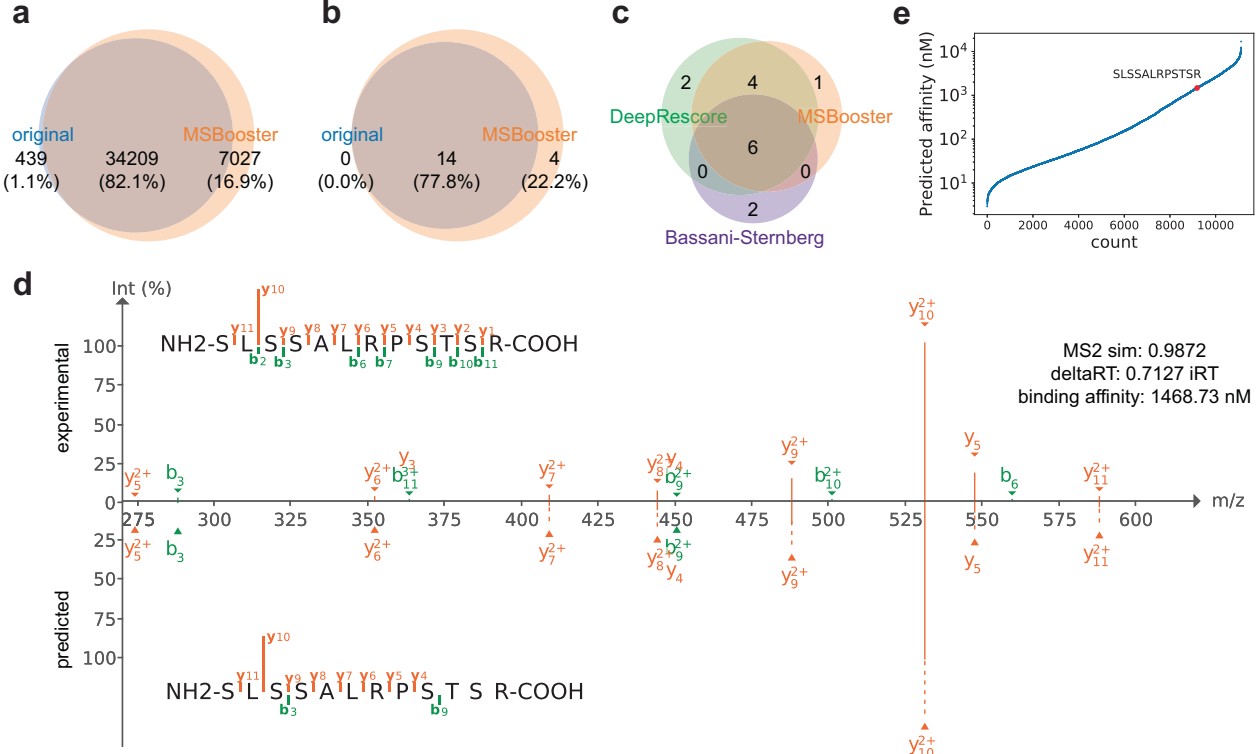

**Fig. 3 | Neoantigen discovery in melanoma tissue. a** and **b** Venn diagrams of peptides identified without (original) and with MSBooster. These peptides are categorized as canonical peptides from the reference database (**a**) or noncanonical neoantigens derived from mutations detected by exome sequencing (**b**). **c** Venn diagram of neoantigens proposed in DeepRescore[65], or our study with MSBooster. Peptides in **a** and **b** were of lengths 7–25, while peptides in **c** were filtered between lengths 8 and 12. **d** PDV visualization of experimental and DIA-NN predicted spectra. y1, y2, b1, and b2 ion intensities are not predicted by DIA-NN and are therefore excluded from visualization. **e** NetMHCpan 4.1 binding affinities of peptides predicted to bind A*03:01. The newly detected peptide SLSSALRPSTSR is shown in red. Source data are provided as a Source Data file.

identifications increased from 14 to 18 (Fig. 3b, Supplementary Data 2). The newly reported neoantigens are credible for multiple reasons. First, three of the peptides (RTYSLSSALR, SLSSALRPSTSR, and SYVTTSTRTYSLSSALRPSTSRS) contain the same single amino acid variant VIM^G41S as six other neoantigens already reported without MSBooster. Furthermore, the 14 original neoantigens had an average MS2 similarity of 0.94 and an average delta RT of 4.1, compared to the 4 new neoantigens with 0.92 average MS2 similarity and 1.4 average delta RT. We also compared our identified neoantigens to those reported in prior analyses of the same data[49,65] (Fig. 3c). Consistent with prior work, we only considered length 8–12 peptides here. Both we and DeepRescore[49] rejected two peptides reported in the original Bassani-Sternberg study[65]—ASWVVPIDIK, which MSFragger did not report, and GRTGAGKSFL (MS2 similarity: 0.81, delta RT: 8.77), which did not pass 1% FDR. Similarly, DeepRescore suggested two peptides that MSBooster did not—DVFPEGTRVGL, which MSFragger did not report, and RLFLGLAIK (MS2 similarity: 0.74, delta RT: 4.12), which did not pass 1% FDR (Supplementary Data 2). MSBooster reported one unique peptide in the allowed length range, SLSSALRPSTSR. The best spectral similarity across all PSMs of this peptide was 0.9872, and the lowest delta RT was 0.7127 iRT (Fig. 3d). Its predicted binding affinity to one of Mel15's alleles A*03:01 was 1468.73 nM, designating it as a weak binder (Fig. 3e).

Previous studies have verified neoantigens by synthesizing them and comparing spectra from the original and synthetic datasets. Comparison of the experimental and predicted spectra functions as a similar quality control measure. To enable researchers to manually verify PSMs, we have incorporated PDV[56] in FragPipe (creating FragPipe-PDV viewer) for visualization of experimental spectra. In addition, we have added support for loading spectral predictions from MSBooster to FragPipe-PDV, enabling the generation of mirror plots to compare experimental and predicted spectra. For the peptide SLSSALRPSTSR, there is high concordance between the spectra, especially with the strong y10^2+ ion (Fig. 3d). Manual comparison of the spectra further corroborates that neoantigens proposed by MSBooster are likely true peptides.

**Direct identification from DIA data**

DIA offers the benefit of monitoring all precursors (within the specified mass range, e.g. 400–1200 Da) and their fragments across retention time, thereby avoiding the stochasticity of DDA which can only produce MS/MS scans for a limited number of precursors. We extended MSBooster to rescoring peptide identifications from DIA data. In FragPipe, peptide identification from DIA data can be performed in two ways: (1) with MSFragger-DIA, which identifies peptides from DIA MS/MS scans by direct database searching; and (2) by first processing the DIA MS files using DIA-Umpire[54] to extract pseudo-MS/MS spectra, followed by searching with MSFragger as regular DDA data. We tested both approaches on a dataset of six melanoma cell lines[66]. MS/MS data were acquired on an Orbitrap Fusion Lumos Tribrid (Thermo Fisher Scientific) mass spectrometer with HCD fragmentation. Using MSFragger-DIA, MSBooster features increased peptide and protein identifications by 16.6% and 8.9%, respectively (Fig. 4a, b, Supplementary Data 1). Using the DIA-Umpire-based workflow, the number of peptide and protein identifications increased by 16.6% and 9.0%, respectively (Supplementary Fig. 5, Supplementary Data 1).

The benefit of rescoring MSFragger-DIA results with MSBooster applies not only to the top-scoring PSMs but also to lower-ranking

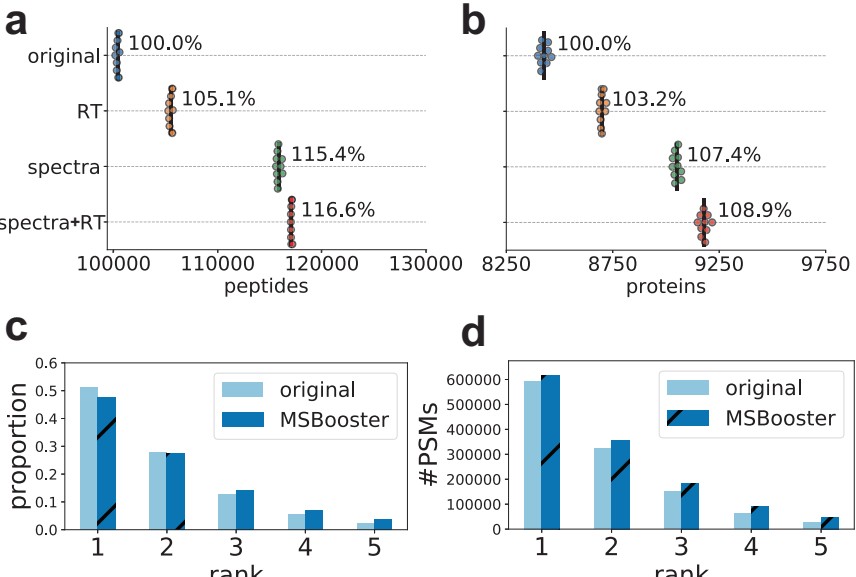

**Fig. 4 | Melanoma DIA rescoring with MSFragger-DIA. a** and **b** Swarmplots of the number of peptides (**a**) or proteins (**b**) reported at 1% FDR. **c** and **d** The proportion of PSMs (**c**) or total number of PSMs (**d**) from each of the five ranks reported. The darker, diagonally dashed bars represent results after spectral and RT rescoring, while the lighter, solid bars represent the results without using deep learning features. Source data are provided as a Source Data file.

PSMs. By default, MSFragger-DIA reports up to 5 PSMs per MS/MS scan. While the initial MSFragger rankings were based on hyperscore, other features provided orthogonal information (Supplementary Fig. 2b) and helped rescue true PSMs with lower hyperscores. With MSBooster, while the total number of PSMs passing the 1% FDR increases across all ranks, a higher proportion of accepted PSMs are from ranks 3 and below, while the relative proportion from ranks 1 and 2 decreases (Fig. 4c, d). MSBooster effectively rescues those lower-ranking PSMs that display characteristics that indicate higher confidence in a true positive PSM.

## Single-cell proteomics

Single-cell proteomics provides a view of the proteomes of individual cells. The lower level of maturity of technological platforms, along with the increased stochasticity of peptide identification due to cell-to-cell variability, make single-cell proteomics another promising area for DL-based PSM rescoring. We tested MSBooster on single-cell data from the nanoPOTS platform[67] generated using an Orbitrap Fusion Lumos Tribrid instrument with HCD fragmentation. Briefly, we analyzed the data obtained from 1, 3, 10, or 50 cells. Single-cell MS/MS spectra differ from bulk-cell spectra in terms of the number of fragments matched and the degree of fragment ion intensity suppression[68]. When looking at the scores of top target PSMs with an increasing number of cells from 1 to 50, we found a trend (Fig. 5a) that with more cells there was a gradual increase in the median spectral similarity among confidently identified target PSMs (i.e., PSMs with expectation values "e-values" lower than the lowest decoy PSM e-value, see the "Methods" section). As a reference, bulk secretome data obtained from An et al.[69], also generated on an Orbitrap Fusion Lumos instrument, demonstrated a higher median spectral similarity score. With respect to RT values, there was a decrease in the median RT difference between 1 and 3 cells, due to an insufficient number of PSMs for optimal RT calibration in MSBooster with one cell only (Fig. 5b). However, the median RT difference did not decrease past 3 cells, because the RT difference should not change once there are sufficient PSMs for RT calibration. The bulk cell RT score distribution was excluded from the comparison because the RT score depends on the individual LC set up. Despite the increasing concordance between experimental and predicted values

with increasing numbers of cells, we did not notice a monotonic relationship between the cell number and MSBooster performance (Fig. 5c, d). The single-cell (1-cell) experiment gained 4.7% and 2.8% peptide and proteins, respectively, with spectral and RT rescoring (Supplementary Data 1). The few-cell data (3, 10, and 50 cells) gained up to 10.6% peptides (in the 50-cell dataset) and 10.6% proteins (in the 3-cell dataset). The RT feature outperformed the spectral feature in many instances.

Because single-cell proteomics methods are being rapidly developed and modified, we tested another dataset produced on a Q Exactive MS with Orbitrap mass analyzer with a different sample processing protocol (DISCO)[45] and HCD fragmentation to see whether MSBooster performance was consistent between different single-cell protocols. Data from 1 and 5 cells were available. While we see similar trends of increasing median spectral similarity and decreasing median RT difference with an increasing number of cells, the median spectral similarity is already above 0.95 for single cells in these data (Fig. 5e, f). In comparison, even 50 cells in the nanoPOTS data had a median spectral similarity below 0.95 (Fig. 5a). We also noted significant differences in the decoy PSMs' spectral similarity distributions (median of 0.49 vs. 0.31, nanoPOTS vs. DISCO) and differing numbers of PSMs reported per replicate (mean of 2924 vs. 19,878, nanoPOTS vs. DISCO). We can see the effect of the higher similarity in the DISCO dataset reflected in the number of peptide and protein identifications achieved with rescoring (Fig. 5g, h, Supplementary Data 1). In this dataset, the spectral similarity feature always outperformed the RT feature. The DISCO dataset experienced greater gains in peptide and protein identifications compared to the nanoPOTS dataset at the single-cell level, possibly in part due to the greater spectral similarity between the DISCO experimental spectra and predicted spectra.

## timsTOF PASEF data with ion mobility separation

Next, we evaluated the performance of MSBooster on data from a HeLa tryptic digest standard analyzed using parallel accumulation–serial fragmentation (PASEF) on a timsTOF Pro (Bruker) mass spectrometer[70], which couples trapped ion mobility spectrometry (TIMS) to a time-of-flight (TOF) detector. Precursors were fragmented with collision-induced dissociation (CID), with the amount of collision

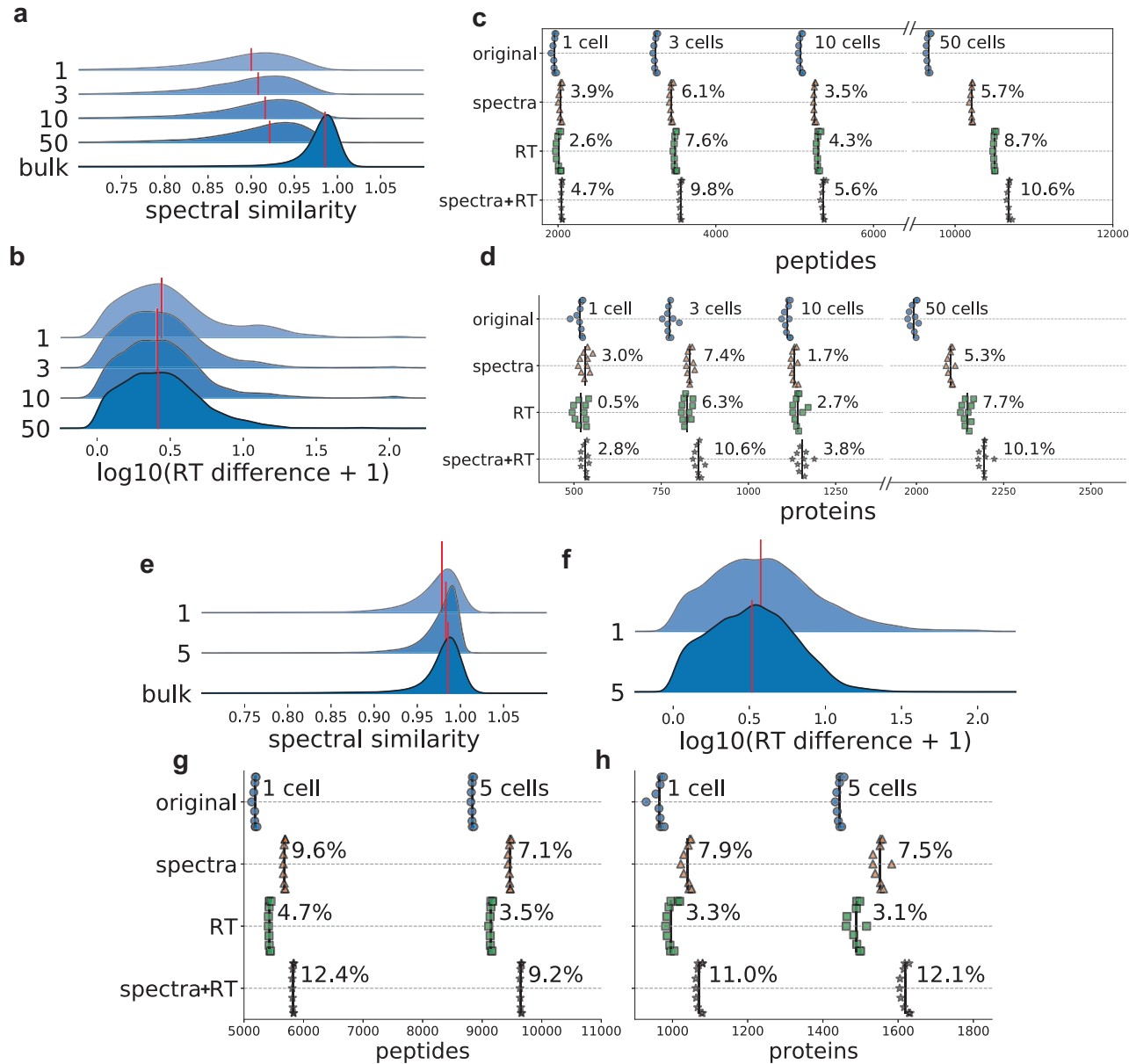

**Fig. 5 | Single-cell rescoring. a–d** Results for nanoPOTS data from Williams et al.[67]. **a, b** Ridge plots showing the distribution of the spectral (**a**) and RT (**b**) feature scores of confident target PSMs for different numbers of cells. The red line indicates the median value. The bulk cell sample is from PXD026436, produced on an Orbitrap Fusion Lumos[69]. The RT feature was log normalized for better visualization. **c** and **d** Swarmplots of the number of reported peptides (**c**) and proteins (**d**) when using different features for Percolator rescoring. **e–h** are the same as (**a–d**), but for the DISCO data from Lamanna et al.[45]. Source data are provided as a Source Data file.

energy as a function of ion mobility. Using both spectral and RT features, we achieved 3.9% and 2.7% increases in peptide and protein identification, respectively (Fig. 6a, b, Supplementary Data 1). While this seems to be a minor increase, it highlights that FragPipe's default workflow for conventional tryptic searches performs well even without DL-based rescoring. In this case, giving Percolator only the hyperscore feature can recover most peptides and proteins without the help of the other features reported by MSFragger and MSBooster (Fig. 6a, b).

Ion mobility (IM) is an additional method for separating precursors prior to MS/MS sequencing. As such, DL models have been extended to predict ion mobility or related collisional cross-section values[38,43] for peptide ions. To assess the utility of predicted IM for PSM rescoring, we ran MSBooster with IM features analogous to its RT features (see Supplementary Note 1). We observed a negligible increase in the number of identified peptides and proteins, below 0.5%

(Fig. 6a, b, Supplementary Data 1), with the addition of the IM score. The weakness of the IM features may be explained by the high dependence of the IM on the precursor mass and charge. Because decoy PSMs still have the same charge and similar mass as the unknown true target precursor, predictions of their inverse ion mobility (1/$K0$) values are still highly correlated with the experimental value (Fig. 6c, d; Supplementary Fig. 6). There is not as much of a spread for IM as for RT prediction (Fig. 6e, f). Overall, while the target PSMs showed a slightly different distribution of IM feature scores from the decoy PSMs (Supplementary Fig. 7e), it was not sufficient to warrant their use in MSBooster.

## Multiple correlated features
Because MSBooster can calculate several variants of spectral, RT, and IM features, we evaluated whether there was value in using

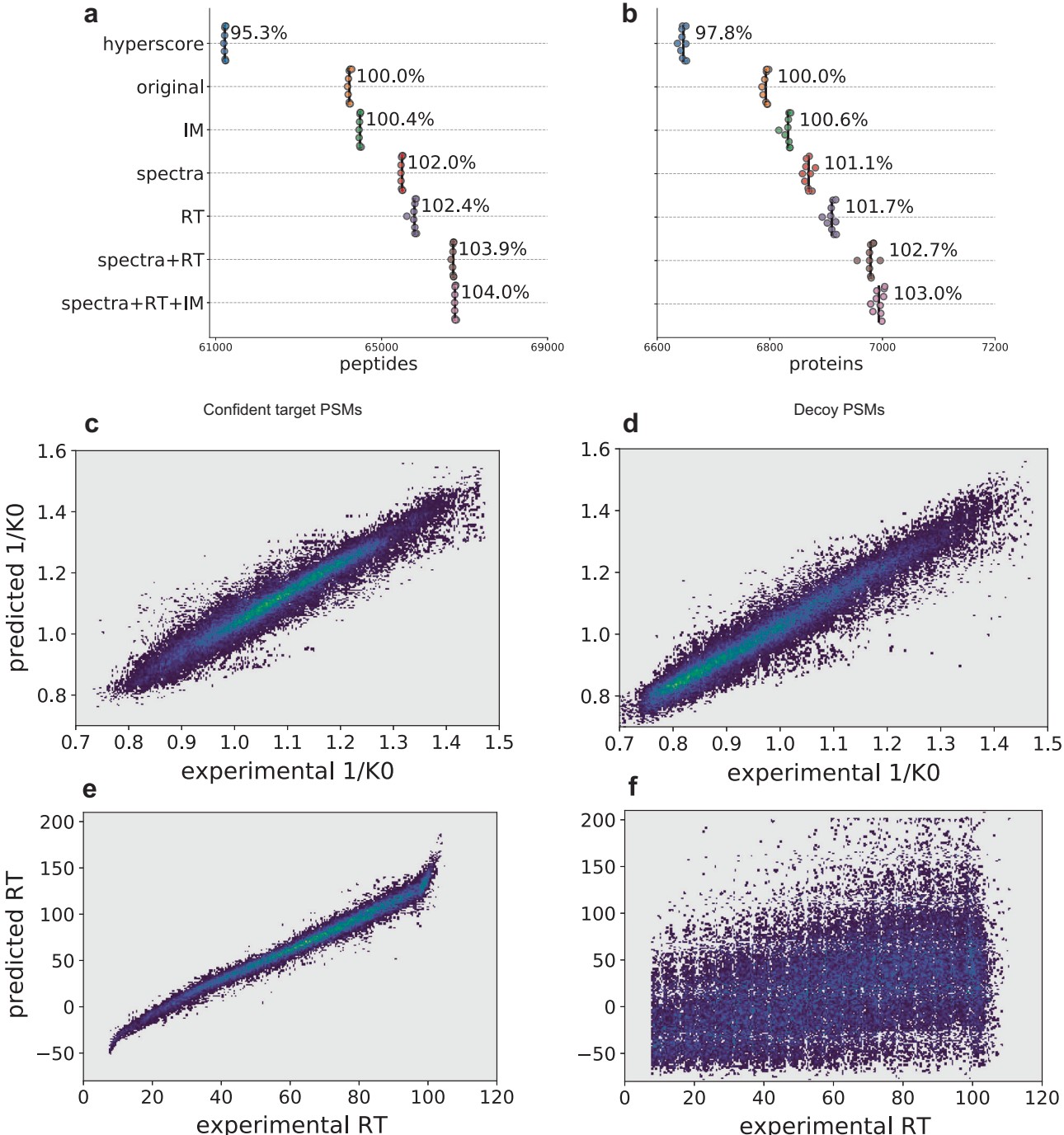

**Fig. 6 | timsTOF HeLa rescoring. a** and **b** Swarmplot of peptides (**a**) and proteins (**b**) reported at 1% FDR. **c**–**f** Scatter density plots showing the relationships between DIA-NN predicted and experimental IM (**c**, **d**) and RT (**e**, **f**) values in seconds for peptides with charges 2 and above. Confident target PSMs are shown in **c** and **e**, decoy PSMs in (**d** and **f**). The brighter colors correspond to higher densities of PSMs. Source data are provided as a Source Data file.

multiple correlated features for PSM rescoring (Supplementary Fig. 8; all tested features are described in the Supplementary Note). This idea was spurred by the finding that even correlated features may not be truly redundant and may work well to provide better separation between classes[71]. For all datasets, we annotated Percolator input files with either single features ("spectra + RT" and "spectra + RT + IM") or all available features listed in the Supplementary Note ("multiple spectra + RT" and "multiple spectra + RT + IM"). In most analyses, the use of multiple correlated features resulted in a minor (<1%) increase in identification numbers. Occasionally, the numbers decreased by an equally small amount

(Supplementary Fig. 8). To investigate this further, we returned to the HLA immunopeptidome dataset. With the use of multiple correlated features, compared to using only a single feature for each feature type (as in Fig. 2), 165 peptides were lost, and 184 were gained. Both groups of lost and gained HLA peptides had comparable binding rates (123/165 lost were binders vs. 134/184 gained were binders; $\chi^2$(1, $N = 349$) = 0.1325, $p = 0.716876$, chi-square test) and similar sequence motifs (Supplementary Fig. 9). However, because it is difficult to rule out all scenarios where using correlated features may be beneficial, we provide an option for FragPipe users to enable their use.

## Discussion

MSBooster is a new addition to the FragPipe computational platform that provides a boost in the number of identified PSM, peptides, and proteins by generating deep learning-based features for PSM rescoring with Percolator. It automatically runs a DL model to acquire predictions for RT, IM, and MS/MS spectra, and generates features based on these predictions to expand the list of scores for each PSM that are useful for discriminating between true and false matches. Furthermore, a notable benefit of MSBooster is that the whole peptide library does not need to be predicted; only high-scoring candidates identified by MSFragger are evaluated using MSBooster, saving a large amount of time, especially for nonspecific searches. Importantly, MSBooster is fully incorporated into FragPipe: a single checkbox enables deep learning prediction and the addition of new features for Percolator rescoring, allowing even inexperienced users to immediately see improved peptide and protein identifications.

We evaluated the improvements provided by MSBooster in various experimental workflows available in FragPipe. We observed robust gains across applications, especially in analyses exhibiting a large search space, such as HLA immunopeptidomics, or multiple peptide candidates per MS/MS scan, such as direct identification from DIA data searches. We found that MSBooster has similar performance to DeepRescore[49] when rescoring a patient-specific melanoma dataset, while also proposing unique neoantigens. MSBooster in FragPipe also outperforms MaxQuant rescoring when both have access to a Prosit-predicted library. Several factors may contribute to MSBooster performance, including MS2 spectral quality, the deviation between the experimental data acquisition parameters and those of the training data for the prediction models, and the number of PSMs available for rescoring. We present general guidelines for what level of gains are expected from multiple popular applications of MSBooster, but an in-depth analysis of how each of these characteristics of the data is outside of the scope of this study.

Interestingly, we observed only a marginal impact of adding IM features when analyzing standard HeLa tryptic digest timsTOF PASEF data. However, this does not mean that IM features will not be useful in other scenarios. IM may improve the resolution of peptidoforms with isobaric modifications (e.g., in glycopeptide identification workflows[72]) or assist with PTM site localization[73,74]. The minimal strength of IM features observed in this work may also be due to insufficient prediction accuracy, suboptimal feature curation in MSBooster, or the limitations of a linear SVM model in Percolator. Thus, more flexible models[75] for PSM rescoring could be investigated. In addition, we compared our IM feature to using only raw $1/K0$ and charge values for rescoring and found that the latter had better performance. We present the results in Supplementary Note 2 and Supplementary Figs. 10, 11.

Future work will focus on making MSBooster more flexible. First, as FragPipe and its constituent tools evolve, MSBooster can adapt too. For example, tools such as ionbot and CHIMERYS can report multiple PSMs per spectrum, potentially allowing consideration of more candidate sequences or co-fragmenting peptides[76,77]. MSFragger could be optimized for identification of co-fragmenting precursors present in DDA data, and MSBooster could be adapted to rescore multiple peptides reported for chimeric spectra, in a similar fashion to rescoring multiple ranks in DIA data.

Second, we plan to extend MSBooster's flexibility via the availability of a standalone command line version to be incorporated into various pipelines outside of FragPipe. This would be applicable for users interested in de novo sequencing[78–80] or using MSBooster in conjunction with other PSM rescoring tools besides Percolator[75,81]. Also, as these interests arise, MSBooster can be adapted to work with PSM table formats besides Percolator pin files, such as pepXML files.

Third, we can extend MSBooster with respect to which prediction model is used. We have already shown Prosit to be compatible with standalone MSBooster in our HLA example, and we plan to include an option in future FragPipe releases for users to easily leverage Prosit predictions. Another example is PredFull[82], a full spectrum prediction model that predicts intensities for every $m/z$ bin, rather than predicting specific ion types such as y- and b-ions. Therefore, it may be able to report internal fragment ion intensities, which could provide more rescoring information for HLA peptides lacking the basic C-terminal residues—common to tryptic peptides—that help to create a strong y-ion series. Other scenarios where non-y/b ions are relevant include rescoring ETD MS/MS spectra, or spectra produced by peptides with PTMs that incur neutral losses. Support for diverse prediction models within MSBooster will be particularly useful for studying PTMs, where models such as pDeep2[41], MS2PIP[36], and DeepLC[83] are expected to perform better than simply shifting fragment ions or using the same RT for both modified and unmodified peptides. The need for models supporting diverse PTMs is evident from the penalty incurred against peptides with pyro-glutamation from Q (Supplementary Fig. 4). The increased use of labeled quantification with tandem mass tags (TMT) has also allowed DL models to be trained for this purpose. Finally, transfer learning and fine-tuning implemented in pDeep3[39] and AlphaPeptDeep[40] may help to create models better suited for different scenarios. For example, while single and bulk cell spectra appear similar on a timsTOF Pro instrument[84], they appear different enough on an Orbitrap instrument that one may consider a model tuned for single cells[68]. Different fragmentation mechanisms, mass spectrometers, and collision energy settings also impact MS/MS spectra. These factors are not currently considered by DIA-NN peptide prediction, but they can have noticeable effects on spectra. We expect that MSBooster's incorporation in FragPipe will allow for ease of customization of which prediction model it is coupled with.

## Methods

### MSBooster workflow

Workflows with and without MSBooster are depicted in Fig. 1. In DDA experiments, MS/MS spectra are searched using MSFragger. For peptide identification from DIA data, either full MS/MS spectra are searched using MSFragger-DIA, or DIA-Umpire extracted pseudo-MS/MS spectra are searched using MSFragger as conventional DDA files. MSFragger produces pepXML and pin files, and the latter is used as input into MSBooster. The pepXML files are not used by MSBooster but are necessary for converting the Percolator output files into pepXML files for subsequent protein inference analysis using ProteinProphet. To obtain a list of peptides for DL model prediction, MSBooster iterates through all pin files to obtain all target and decoy peptides matched to at least one PSM. Peptides with the same sequence but different PTMs and/or charges are treated as different peptides. This list is then passed to a DL model to obtain a prediction file. This strategy is significantly faster than predicting the entire in-silico digested proteome, as only peptides reported as potential hits by MSFragger (top-ranking peptides per spectrum) are submitted for prediction. Peptides with PTM(s) not supported by the DL model rely on predictions for the peptide without the unsupported PTM(s). MSBooster adds a shift in fragment $m/z$ to accommodate the new PTMs, but the fragment intensities remain the same.

The core of MSBooster is the feature calculation step. First, the predictions from the DL model are loaded. Then pairs of mzML (or MGF) and pin files are sequentially loaded for processing and DL-extended PSM feature table generation. For DIA data, because multiple peptides may contribute to a single MS/MS scan, the experimental spectra are revised after each PSM has its features calculated. Borrowing from MSFragger-DIA, the highest intensity experimental MS/MS peak within the fragment error tolerance of the reported predicted fragment is removed from the experimental spectrum. The predicted spectra of lower-ranking PSM peptides can no longer have their fragments matched to these removed peaks because using the same MS/

MS peaks for multiple PSMs from the same scan can lead to spurious hits. Once all PSMs from a single-pin file are loaded, RT and IM calibrations are performed. In the final step, MSBooster iterates through the pin file row by row and calculates and adds the desired features. This process is repeated until all pin files have DL features calculated and added. Multiple features were tested and are discussed below. The list of all available features is described in Supplementary Note 1.

### Determination of the best features

Several metrics exist for calculating the similarities between experimental and predicted spectra. Although cosine similarity is commonly used, several features were tested to determine which metrics could provide the greatest gains in the identification numbers (Supplementary Fig. 12). Percolator is non-deterministic because of the random splitting of PSMs for training and testing, which can be controlled with a random seed. Thus, Percolator was run ten times for each feature, and the number of peptides reported after Philosopher filtering was counted. For spectral similarity features, the greatest boosts were consistently obtained with "unweighted spectral entropy"[85]. For the RT features, "delta RT loess" tended to do the best. Interestingly, "delta RT loess normalized" performed better when there were a small number of cells in the nanoPOTS data[67] (Supplementary Fig. 12e–h). We tested a linear regression feature for RT calibration, "delta RT linear", on the HLA and 50 cell datasets (Supplementary Fig. 12a). While it performed similarly to "delta RT loess normalized" on the HLA dataset, we found that it may be performed sub-optimally for the 50-cell data, where there exists a non-linear relationship between the experimental and predicted RT scales (Supplementary Fig. 13). For the IM features, the "IM probability uniform prior" feature performed the best. The distributions of each score for all PSMs, targets and decoys, in the different datasets are shown in Supplementary Fig. 7.

### MSFragger search and FDR control

Database searches were performed using MSFragger v3.4 in FragPipe v17.2 with Philosopher v4.1.1. For neoantigen detection, MSFragger v3.7, FragPipe v19.2, and Philosopher v5.0.0 were used for visualization of spectra with FragPipe-PDV viewer[56]. All searches used a UniProt fasta from March 18, 2022, except for the neoantigen search, which used a fasta with both canonical and Mel15-specific protein sequences derived from exome sequencing; this fasta was the same used by Li et al.[49] and generated by NeoFlow[86]. The workflows used for each dataset are as follows: HLA immunopeptidome[57] (nonspecific-HLA-C57 workflow); melanoma neoantigen[65] (nonspecific-HLA with carbamidomethylated cysteine added as a variable modification); melanoma DIA data[66] with MSFragger-DIA (DIA_SpecLib_Quant) and with DIA-Umpire (DIA_DIA-Umpire_SpecLib_Quant); HeLa timsTOF[70], single-cell proteomics with nanoPOTS[67] or DISCO[45], and secretome[69] (Default). All workflows included oxidation of methionine and N-terminal acetylation as variable modifications. The workflows besides Default also included pyroglutamation from glutamine and glutamic acid. The HLA workflow had carbamidomethylated cysteine as a fixed modification with the mass difference between cysteinylation and carbamidomethylation (61.98 Da) as a variable modification; the neoantigen workflow included both carbamidomethylation and cysteinylation as variable modifications. A maximum of three variable modifications was allowed. Peptide length was set to 7–25 for nonspecific workflows and 7–50 for all others. All workflows used 20 ppm for precursor and fragment error tolerance, with mass calibration and parameter optimization enabled. MSBooster, Percolator, ProteinProphet, and Philosopher were enabled. The HLA workflow was revised to add "−mods M:15.9949" to the Philosopher filter to perform group-specific FDR estimation[87] using the following three categories: unmodified peptides, peptides with oxidized M only, and peptides with any other modification. The nanoPOTS data were analyzed in separate experiments based on the number of cells (1, 3, 10, or 50). Peptide and protein identifications

reported are at 1% FDR unless otherwise noted. We attempted to calculate FDR specifically for nonreference targets/decoys in our neoantigen dataset as suggested by Nesvizhskii et al.[88]; however, too few nonreference decoys were reported to accurately calculate a group-specific FDR. Therefore, the numbers reported are those nonreference target peptides in the original peptide.tsv files.

### MaxQuant search and FDR control

MaxQuant v2.1.0.0[9] was used to search the HLA immunopeptidome data[57]. Search tolerance was 20 ppm. For the MaxQuant only search, oxidation of methionine, n-terminal acetylation, pyroglutamation of glutamine and glutamic acid, and cysteinylation minus carbamidomethylation of cysteine were specified as variable modifications. Carbamidomethylation of cysteine was specified as a fixed modification. FDR at all levels was set to 0.01. For the MaxQuant search to be used for Prosit rescoring because the only PTMs supported by the base non-TMT Prosit model are carbamidomethylation of cysteine and oxidation of methionine, the former is set as the only fixed modification, the latter as the only variable modification. All FDR levels were set to 1, as is required for Prosit rescoring.

### MaxQuant analysis of Sarkizova et al.

Results of the analysis of 92 monoallelic HLA Class I cell lines with MaxQuant performed by Wilhelm et al.[46] were downloaded. For each allele, the number of peptides with non-NA scores was counted and compared before and after Prosit rescoring. The average across all cell lines was calculated. Results are shown in Supplementary Fig. 3a.

### Deep learning predictions

DIA-NN v1.8[42,43] was used to predict RT, IM, and MS/MS spectra because of its speed and ease of execution within FragPipe. DIA-NN reports the top 12 most intense singly and doubly charged b- and y-ions. Predictions were made for each unique combination of peptide sequence, modifications, and charge. DIA-NN v1.8 supports the predictions for peptides with carbamidomethylated cysteine, oxidized methionine, N-terminal acetylation, phosphorylation, and ubiquitination. For other PTMs such as pyro-glutamation, DIA-NN did not adjust MS/MS fragment peak intensities, but MSBooster shifted the peaks to the appropriate $m/z$ values. The RTs and IM values for peptides with unsupported PTMs remained the same as for counterparts without the PTM.

Prosit[46] was used for rescoring of HLA immunopeptidome data[57], both with MaxQuant and MSBooster. When running with MaxQuant output, the "rescoring" pipeline at https://www.proteomicsdb.org/prosit/ was used. The msms.txt from the MaxQuant with FDR = 1 search was used as input, along with the individual RAW files. The size of the unique set of peptides with $q$-values < 0.01 are shown in Supplementary Fig. 3b. When used in combination with MSBooster, a command line version of MSBooster was used to extract peptides from the pin files, analogous to how it is done in FragPipe with DIA-NN. The peptides are formatted into an input file for the "spectral library" pipeline at https://www.proteomicsdb.org/prosit/. For both MaxQuant and MSBooster, the "Prosit_2020_intensity_hcd" and "Prosit_2019_irt" models are used for MS/MS spectral and RT predictions, respectively. The resultant "msp" file from Prosit is read by MSBooster, and spectra for peptides with PTMs besides carbamidomethylated cysteine and oxidized methionine are generated via m/z shifting, analogous to how it is done when used in conjunction with DIA-NN.

### Spectral similarity calculation

To calculate the spectral similarity, the highest intensity fragment ions within the $m/z$ error tolerance of the predicted fragment ions are obtained. Therefore, similarity calculations are performed using vectors of the same length. If no peak is detected in the experimental spectrum within the $m/z$ error tolerance of the predicted peak, the

experimental vector is assigned a 0 at that position. Predicted and matched fragment ions from the experimental MS/MS spectra were normalized before similarity calculation (see Supplementary Note 1).

## Retention time and ion mobility calibration
Local regression (LOESS) is used to calibrate the experimental to the predicted RT and IM values, followed by monotonic regression. A different ion mobility model is trained for each charge. The resultant model maps each experimental RT to a value on the predicted RT scale; a calibrated RT value is the experimental RT mapped to the predicted scale. To train the regression models, a subset of PSMs (5000, by default) with expectation values below a preset threshold (10e−3.5) is used. This threshold was chosen from the observation that of the various datasets MSBooster was tested on, no decoys were detected below that expectation value. If between 50 and 5000 PSMs have sufficiently low expectation values, that number of PSMs is used. If fewer than 50 target PSMs with sufficiently low expectation values are available, linear regression is performed instead. For both DDA and DIA, only rank 1 PSM is considered for the regression. The bandwidth by default is set to 0.05 for RT and 0.1 for IM. To calculate the difference between the predicted and experimental RT, the experimental RT is first calibrated to the predicted scale using the regression model, followed by calculating the difference between the calibrated RT for that MS/MS scan and the predicted RT for that peptide. The same is performed for the IM.

## Kernel density estimation of predicted retention time and ion mobility distributions
The following discussion uses RT, but the same applies to IM. Empirical distributions of predicted RT/IM values were generated using statistical machine intelligence and learning engine (Smile) implementation of kernel density estimation (KDE) with a Gaussian kernel (https://haifengl.github.io/api/java/smile/stat/distribution/KernelDensity.html). The bandwidth of the kernel is estimated by Silverman's rule of thumb, as implemented by Smile. Briefly, the algorithm works by replacing each point in the distribution with a Gaussian curve of equal amplitude, then summing all the individual curves into one total distribution that is a smoothed version of the empirical distribution. The empirical range is divided into equally sized bins (widths of 1 min for RT, 0.01 1/$K0$ units for IM, by default). For each PSM, its predicted RT is placed into a bin with all predicted RTs of PSMs from the same experimentally observed RT minute. The number of times its RT value is added to the bin is weighted by its expectation value; that is, a higher-confidence PSM with a low expectation value will have its predicted RT added to the bin more times than a lower-confidence PSM with a high expectation value. After all predicted RTs are placed in their respective bins, KDE is used to generate empirical distributions. These distributions can be used to estimate the probability of having a PSM with a predicted RT value given its experimental RT and are not subject to the monotonic constraint of the LOESS model. For example, to get the "RT probability" value for a peptide scanned in the 60th minute, the KDE distribution from the 60th-minute bin, $D$, is queried. If the peptide has a predicted RT $R_P$, then the reported value for this feature is the KDE estimated probability ($P_E$) of having predicted RT $R_P$ from the 60th-minute bin's probability distribution, $D$. The same procedure is applied to the IM to generate distributions of the predicted IM, separating the PSMs by charge state. The features from MSBooster that use these probabilities also add a uniform prior distribution to the KDE-generated distribution. This uniform prior helps to dampen the effects of bins with fewer entries. For example, if an experimental RT bin contains a single PSM, not using a uniform prior would result in an artificially high probability for that PSM. The uniform prior is decided by sorting

the RT bins in ascending order by the number of PSMs contained in each. A bin is chosen based on a preset percentile (10th percentile, by default). The number of PSMs $U$ at the RT bin at this specific percentile is chosen. The uniform prior probability $P_U$ is distributed equally across the predicted RT range. If a PSM with empirical probability $P_E$ is placed in a bin that contains E PSMs, the value of the "RT probability uniform prior" feature is described by Eq. (1):

$$\frac{P_U * U}{U + E} + \frac{P_E * E}{U + E} \tag{1}$$

## HLA motif analysis
Swarmplots and Venn diagrams were generated with no filtering of peptides based on length. Before using peptides in GibbsCluster 2.0 (https://services.healthtech.dtu.dk/ service.php?GibbsCluster-2.0)[60] or the NetMHC software (https://services.healthtech.dtu.dk/ service.php?NetMHC-4.0; https://services.healthtech.dtu.dk/services/NetMHCpan-4.1/)[62], they were filtered to be between lengths 7 and 12 for Klaeger et al.[57], or between lengths 8 and 12 for Bassani-Sternberg et al.[65]. Position weight matrices were generated using GibbsCluster. The binding affinity of the peptides to the A*02:01 MHC was determined using NetMHC 4.0 using the default settings. Similarly, the binding affinity of peptides from Bassani-Sternberg et al.[65] to A*03:01 was determined using NetMHCpan 4.1, which accommodates peptides longer than length 11, such as our detected neoantigen.

## Statistical analysis and figure generation
Figures were generated in Jupyter Notebooks using Python 3.7.6, Anaconda 2020.02, Conda 4.8.2, Joypy 0.2.6, Jupyterlab 1.2.6, Matplotlib 3.1.3, Matplotlib-venn 0.11.7, Numpy 1.18.1, Pandas 1.3.0, and Seaborn 0.10.0. The scatter density plots for Fig. 6 and Supplementary Fig. 6 require a separate Anaconda environment with Python 3.8.3. Statistical tests such as $t$-tests were all two-sided and performed with SciPy 1.4.1.

## Hardware
FragPipe was run and timed using Java 16.0.1. A command-line version of MSBooster was run on a Windows desktop with 12 logical CPU cores (Intel(R) Core™ i7-8700 CPU @ 3.20 GHz) and 32 GB of memory. This was essential for automating the testing with different MSBooster features.

## Reporting summary
Further information on research design is available in the Nature Portfolio Reporting Summary linked to this article.

# Data availability
MS/MS datasets used in this study can be found at the ProteomeXchange Consortium and the PRIDE partner repository[89] or at the MassIVE repository with the following accession codes: HeLa timsTOF DDA PXD010012[70], HLA peptidome MSV000087743[57] [https://massive.ucsd.edu/ProteoSAFe/dataset.jsp?task=a1638beae5d04267a99f92c550c60b34], melanoma neoantigen PXD004894[65], melanoma DIA PXD022992[66], single cell nanoPOTS MSV000085230[67] [https://massive.ucsd.edu/ProteoSAFe/dataset.jsp?task=3013fc11dc4e4b6dae49a244d92854a7], single cell DISCO PXD019958[45] and secretome PXD026436[69]. All MSFragger-produced pepXML, MSBooster-annotated pin, and fasta files are available at https://doi.org/10.5281/zenodo.8034585 and https://doi.org/10.5281/zenodo.7843558. Data used to generate the main and supplementary figures are provided in the Source Data file. MHC allele binding motifs were acquired at the Immune Epitope

Database (https://www.iedb.org/). Source data are provided with this paper.

## Code availability

MSBooster code is available freely and as open source at https://github.com/Nesvilab/MSBooster.

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

## Acknowledgements

This work was supported in part by National Institutes of Health grants R01-GM-094231, U24-CA210967, and U24-CA271037 received by K.L.Y., F.Y., G.C.T., K.L., and A.I.N.; and the Proteogenomics of Cancer Training Program 5T32-CA140044-12 received by K.L.Y. and A.I.N.

## Author contributions

K.L.Y. developed the algorithm, wrote the software, and analyzed the results. F.Y. assisted with the algorithm and software development. F.Y. and G.C.T. provided support for integration of MSBooster in FragPipe. K.L. provided support for integration of MSBooster in FragPipe-PDV. V.D. and M.R. provided resources pertaining to DIA-NN. K.L.Y. and A.I.N. wrote the manuscript. A.I.N. conceived and supervised the study.

## Competing interests

The authors declare no competing interests.
