## [Peer Review File · Nature Communications]

REVIEWER COMMENTS

Reviewer #1 (Remarks to the Author):

The authors present work to improve identifications rates of peptides through the incorporation of new parameters predicted from deep-learning models. The authors go on to show the utility of new parameters related to chromatographic retention time, ion mobility, and spectral quality. By including these new features the authors claim large improvement in total peptide (and protein) identifications. Interestingly, the authors show that some of their largest benefits are for some of the most challenging proteomics analysis types including immunopeptidomics and single cell proteomics. My only concerns with the manuscript as is relate to novelty and extensibility of the methods, which I believe would require some revision prior to final publication.

Concerns and Questions

1. It would be helpful to clarify exactly what MSBooster is doing algorithmically for this work. My impression is that MSBooster is solely for estimating 2-3 additional parameters. The actual prediction of these parameters (e.g., RT) seem to be coming from DIA-NN, and the filtering is done post hoc with Percolator. This could be added to Figure 1 for example where a better illustration of “Feature Calculation” would go a long way for interpretation.
2. Following from the above, it would be helpful to know how the “basic” features would do to improve the identification rates. For example, would raw deltaRT between DIA-NN predictions and experimental data enable similar discrimination of good hits? If this was done, can the authors add this to a figure such as Figure 2A?
3. I eventually found it, but it’s not clear from the paper or fragpipe’s website where the standalone MSBooster tool is.
4. I would highly recommend enabling output beyond the `pin` file. There are other means to filter PSMs beyond percolator (and new methods that folks will try in the future) and these would need a flatfile type of some sort that is more generic.
5. For the IM data, what input can be used? Such as, would DMS/FAIMS data be enabled instead of tims?
6. “...this may simply reflect the moderate performance of MaxQuant in nonspecific searches...” The authors should include a comparison to MaxQuant rather than a reference to make the comparison “fair” here.
7. Did the DIA data have IM features? If not have the authors tried running these comparisons using IM-DIA data?
8. P7, lines 212-214. Can you add the raw numbers in addition to the percentages and note that 80% is a sum of the high/weak binders?

9. "This indicates that Percolator gives little preference to higher-ranking PSMs in DIA data..." Is rank weight masked by other factors? For example, if hyperscore and rank are strongly related. Also, how is this affected by the inclusion of only rank 1 PSMs in the regression (p16, line 506)?

10. "Overall...IM feature scores...not sufficient to warrant their use in MSBooster." From Figure S5b, it appears that IM data and mass to separate decoys with the main group of decoys coming from 1+ precursors. A linear model accounting for both parameters in a single value might be enough to make the IM data more useful and seems worth trying.

11. Similarly, does the use of IM features change the weights for charge?

12. What is the "grey zone"?

13. "...MSBooster can be used to process the MSFragger output for DDA data with multiple PSMs reported per spectrum...". This is potentially very exciting, but if it's claimed then there should be data to back it up. Please add a figure for this.

14. Supplementary Figures. There seems to be an issue with the supplementary figures as they are at low resolution which makes interpretation nearly impossible for some (Figure S2, S6, S7).

Minor Concerns and Questions

- "Eluding" is used multiple times instead of "eluting". E.g., p3, line 54.
- P3, line 62. Please correct "data independent acquisition (DDA)".
- There are a lot of parentheses in the introduction (sometimes two to three in one sentence: p4, lines 82-87) that suggest the sentence structure could be rearranged to make some sections more clear.
- P7, line 200: "...FDR were shared regardless of [whether] MSBooster was used..."

Reviewer #2 (Remarks to the Author):

This is a well written article that proposes a relatively straightforward idea and tests it thoroughly and convincingly. Certainly, the idea that deep learning models can be used to help improve statistical power to detect peptides from mass spectrometry data is not novel. And the approach proposed here -- augmenting the feature vector that goes into Percolator with a collection of scores from a pre-trained deep learning model -- is neither super novel nor profound. But it's also the most straightforward way to modify existing pipelines like MSFragger and, as the authors show here, it works quite well in practice.

The introduction is very nicely written, giving a brief but accurate synopsis of the field and of this work's contribution to it. I liked that the results section concisely covers a diversity of applications, from HLA peptide identification to DIA analysis to single-cell analysis and even timsTOF data. In each case, the analysis shows convincingly that using the deep learning features boosts power to detect peptides, to varying degrees. I have only a few relatively minor critiques.

It's not accurate to say that existing post-processing algorithms "do not incorporate prior knowledge regarding peptide separation coordinates" (line 95). Until recently, Percolator included a specific set of features to model RT (see Moruz JPR 2010). This same paragraph later addresses this point, and even cites the Moruz paper, though it implies incorrectly that only PeptideProphet has incorporated such estimates into its models.

In the description of MSFragger searching, I am guessing that the software includes predefined sets of parameters, referred to here as "workflows." This point needs to be clarified in the text, because this will not be obvious to people who have not used your software before.

The paragraph beginning on line 218 talks about how MSBooster handles PTMs that are not supported by DIA-NN. This is mildly interesting, but I don't think the analysis (Supp Fig 4) directly shows what you claim, namely, that "MSBooster allows re-scoring of any PTM-containing peptide, even if a small penalty is applied in the case of certain PTMs (not yet supported by DIA-NN prediction module)." To me, what would be interesting is information about the numbers of such modifications that are detected with and without the DL features. The analysis in this paragraph really only shows that such PTMs receive a small penalty in score.

The obvious question posed by Figure 3A-D is why 3B/D are included at all. The conclusion (which was already apparent from the MSFragger-DIA paper) is that DIA-Umpire does not work as well as MSFragger-DIA. Given this, I would recommend dropping 3B/D or, at best, moving them to a supplementary figure.

line 504: Please tell us what "preset threshold" is used in this work.

line 525: Also tell us the weight of the uniform prior.

In the supplemental note, please be more precise about the RT loess term calculation: "some subset of PSMs with lowest expectation values" is not specific enough. Is there a rank threshold involved? Also, please specify how the calibration of experimental to predicted RT is done exactly. For the "RT probability unif prior," specify exactly how the Gaussian smoothing is done, and how the kernel mean is

selected. I don't actually understand this sentence: "A PSM can be rescored by matching its experimental RT to the appropriate bin and calculating the probability from the empirical distribution." Please expand this description.

Minor:

In Figure 1, it seems strange to me that there is an arrow from "PSM feature tables" to Percolator. Don't the "Extended PSM feature tables" contain all the features?

line 185: I assume that these results are at a 1% peptide-level FDR, but I think this should be stated explicitly.

line 200-201: Missing word ("whether")

Line 334: I think you should define "1/K0" the first time it is used.

line 431: searches -> searched

Reviewer #3 (Remarks to the Author):

The manuscript, "MSBooster: Improving Peptide Identification Rates using Deep Learning-Based Features" by Yang et al., reported the advantage of combining the deep learning (DL) module of "DIA-NN" that was originally reported by the two authors of this paper (Demichev, V., et al., Nature Methods, 2020) and "MSFragger" that was reported by the last author of this paper (Kong, A.T., et al., Nature Methods, 2017). The authors used their knowledge of coding and MS-search to construct DL-assisted "MSBooster" and validate its utility by using existing repository datasets.

Specifically, they used monoallelic immunopeptidomics, DIA proteomics, single-cell proteomics, and timsTOF datasets, showing that RT and spectral data improved peptide identification efficiency. The codes that the authors have already established in the past (DIA-NN and MSFragger) are both excellent, and the challenge of combining the two to further improve identification efficiency is commendable. On the other hand, in light of the academic impact of the journal, it can not be denied that the content of this manuscript lacks the novel findings obtained by combining the MSFragger with DIA-NN. Given the current increasing reports on the construction of similar algorithms and the launch of commercially-

available DL-assisted software, additional validation is still necessary to clarify the advantage of MSBooster.

Major Comments

1. From Figures 2-4, it appears that the improved efficiency of peptide identification is mainly useful for immunopeptidomics (up to 31.4% increase by Spectra+RT similarity features, other datasets limited from a few to 16.6% increase at most). However, validation search by monoallelic sample (only with A*02:01 allotype) seems too simplistic to demonstrate the efficacy of MSBooster.

Immunopeptidomics is highly desired in clinical immunology in which datasets from human specimens are important. This means necessity of immunopeptidomics under full allelic conditions, including up to six HLA motifs. Therefore, it's better to show the results of validation using previously reported actual clinical datasets (e.g., Melanoma sample by Bassani-Sternberg, M. et al. *Nat. Commun.* 7, 13404 (2016), TNBC sample by Ternette, N. et al. *Proteomics* 18, 1700465 (2018), CRC sample by Minegishi, Y. et al. *Commun. Biol.* 18;5(1):831 (2022), etc.) to determine the degree to how much the efficiency of peptide identification can be improved by MSBooster.

Since the DL-assisted search software (Proteome Discoverer 3.0, etc.) as well as the immunopeptide-specific de novo sequencing software (DeeNovo by PEAKs, etc.) have commercially launched, it is necessary to demonstrate and emphasize the scientific advantage of MSBooster. For that, if neoantigens that could not be retrieved by the past search methods could be identified from the reposit datasets, I believe that this manuscript is worthy of publication.

Due to the current upgrowth of reports on the construction of algorithms that claim to improve efficiency of peptide identification, additional clear examples except for immunopeptidomics should be crucial to show the feasibility of what new findings can be retrieved by MSBooster.

2. As authors mentioned by themselves, since the ion mobility works on the MS1 precursor ion separation, the DL algorithm in MS2 fragment ions does little in improved identification. So, the effect of using DL-assisted search by in ion mobility does not seem to be a matter mentioning in result section at least in this manuscript.

The result section of timsTOF PASEF data with ion mobility can be briefly mentioned in discussion section (Tried to introduce IM factor into DL features but didn't work because IM works on MS1 separation, etc.) and can replace the Figure 5 into Supplementary information.

3. The obtained MS2 spectra are different by distinct collision modes. The detectors of MS2 are also different from mass spectrometer to the other. The ion mobility-assisted MS can be conducted by not only timsTOF but also by the FAIMS-interface. As such, the data acquisition conditions on the MS side can be a rate limiting factor for peptide search by each algorithm.

Therefore, it is informative to clearly mention the details of MS parameters; collision mode (HCD/CID/ETHcD, etc.), MS2 detector (TOF/Orbitrap/Iontrap, etc.), with or without IM options (timsTOF or FAIMS interface, etc.) in each dataset used in the validation study. Then comparative study of which MS conditions can be the most beneficially MSBooster-search can clearly demonstrate the further superiority of this algorithm.

Minor Comments

1. In main text, line 294-296, "Surprisingly, when looking at the different cell numbers separately, we did not notice a clear relationship between the cell number and MSBooster performance (Fig 4c-d)". What does this "a clear relationship" mean? Didn't this simply mean the search by MSBooster for nanoPOTS dataset did not yield increased results corresponding to the increased cell counts? Doesn't this simply mean the MSBooster does not work reliably with datasets that contain too little information, such as single cell proteomic by nanoPOTS data? Or is it because the cellular proteome, which is the primary purpose of single cell analysis, is incompatible with the system of MSBooster's rescoring because of the distinct major protein expressions in each cell types included in the target sample? Wasn't that why the results by MSBooster didn't correspond to the number of cell counts? Or, the authors meant the data acquisition by nanoPOTS is qualitatively poor (or not applicable for MSBooster?) compared to the DISCO method, that was why no clear relationship between the cell number and MSBooster performance? As I asked in the major comment #2, it is important to mention the compatibility of MSBooster with the conditions/parameters of data acquisition by MS-side. Please clarify and explain all these points in the main text.
2. In main text, line 312, "10-15%" is an overstatement. In Figure 4g-h, maximum increase was described as 12.4% (Fig. 4g, spectraRT). This is only 12% when rounded off. Please correct the description accurately according to the data, or avoid overstatement.
3. In Figures 2a, 3a-d, 4c-d, 4g-h and 5a-b, it seems that the statistical evaluation can be feasible for the rate of increase in identification by analysis using MSBooster. Please consider to add the statistical analyses to show the advantage of MSBooster clearly.
4. The resolution of Supplementary figure 2 is too poor to read the embedded-text. Please use the higher resolution, or add readable text in a larger font size.

Reviewer #1 (Remarks to the Author):

The authors present work to improve identifications rates of peptides through the incorporation of new parameters predicted from deep-learning models. The authors go on to show the utility of new parameters related to chromatographic retention time, ion mobility, and spectral quality. By including these new features the authors claim large improvement in total peptide (and protein) identifications. Interestingly, the authors show that some of their largest benefits are for some of the most challenging proteomics analysis types including immunopeptidomics and single cell proteomics. My only concerns with the manuscript as is relate to novelty and extensibility of the methods, which I believe would require some revision prior to final publication.

Response: We thank the reviewer for their insightful summary and comments. We would like to highlight that one of the key novelties of MSBooster is its ease of use, which we have now emphasized in the Discussions section. The integration of MSBooster into FragPipe workflows has already been adopted by many users due to its user-friendly nature. This sets it apart from tools that necessitate external prediction of spectral libraries, such as MaxQuant with ProSight. Our streamlined process facilitates accessibility for researchers with limited computational experience. Additionally, we would like to point out that commercial tools can be expensive for academic users and often require data to be uploaded to the cloud, potentially raising concerns about data security. We have addressed all of the reviewer's comments and provided a point-by-point response in the subsequent sections. We point to the location in the manuscript where changes have been made, as well as including the changes here (in *bold Italic*).

Concerns and Questions

1. It would be helpful to clarify exactly what MSBooster is doing algorithmically for this work. My impression is that MSBooster is solely for estimating 2-3 additional parameters. The actual prediction of these parameters (e.g., RT) seem to be coming from DIA-NN, and the filtering is done post hoc with Percolator. This could be added to Figure 1 for example where a better illustration of "Feature Calculation" would go a long way for interpretation.

Response: Thank you for this insight. We have redone Figure 1 to more clearly highlight the contribution of MSBooster to calculate features comparing predicted and experimental values. We have included the following clarification at line 149-166.

Manuscript changes: Within FragPipe, MSBooster is positioned between MSFragger and Percolator (Fig 1a) and is enabled by default in most FragPipe analysis workflows (see Methods section for details), where a FragPipe workflow is the order in which software are to be executed, along with optimized parameters for each tool. MSBooster's role can be divided into the separate steps of peptide extraction from PSM files, input file formatting for a DL model, feature calculation using observed and predicted peptide properties, and addition of the new features to the PSM files. In a typical workflow, MSFragger performs the database search, and reports the list of PSMs and associated search scores in a "pin" file (Fig 1b). Without MSBooster, these pin files are passed directly to Percolator. When MSBooster is enabled, it extracts the set of peptides reported in the pin file and creates an input file for a DL model, which in turn generates predictions of the physicochemical properties of peptides, namely RT, IM, and/or MS/MS

spectra. Within FragPipe, we chose DIA-NN [42, 43] to predict these properties, as it is already included for DIA quantification. We also show compatibility of Prosit predictions with standalone MSBooster, although this is not yet supported in FragPipe. Importantly, predictions are performed only for the relatively small set of PSM candidate peptides reported by MSFragger, rather than the whole in-silico digested proteome. Thus, predictions can be done for each dataset on-the-fly, without the need for time-consuming full spectral library prediction. MSBooster then generates features based on the agreement between the experimental and predicted values and adds them to the original pin files.

2. Following from the above, it would be helpful to know how the “basic” features would do to improve the identification rates. For example, would raw deltaRT between DIA-NN predictions and experimental data enable similar discrimination of good hits? If this was done, can the authors add this to a figure such as Figure 2A?

Response: Calculating raw deltaRT before calibration will not be optimal due to differences in scale between experimental and predicted RTs. Experimental gradients may range between minutes to hours. The predicted scale is between about -80 to 220 iRT units. If the experimental gradient were 5 minutes (5 RT units), for example, true peptides eluting near the end would be heavily penalized (e.g. $\text{absolute}(220 \text{ iRT units} - 5 \text{ experimental RT units}) = 215$) and false peptides eluting at the end but predicted to elute at the beginning would perform better (e.g. $\text{absolute}(-80 \text{ iRT units} - 5 \text{ experimental RT units}) = 85$).

We tested a simple linear regression feature, deltaRT linear. We have added it to Supplemental Figure 12a to show that deltaRT loess is still superior. We have also added a Supplemental Figure 13 showing the non-linear nature of the experimental-predicted RT relationship; it is especially clear that in this example, precursors eluting near the end are heavily penalized by the linear regression feature, as shown in the inset. We discuss the linear feature briefly in the Methods “Determination of the best features”, line 533-548.

Manuscript changes: We tested a linear regression feature for RT calibration, “delta RT linear”, on the HLA and 50 cell datasets (Supplemental Fig 12a). While it performed similarly to “delta RT loess normalized” on the HLA dataset, we found that it may performed sub-optimally for the 50-cell data, where there exists a non-linear relationship between the experimental and predicted RT scales (Supplemental Fig 13).

3. I eventually found it, but it’s not clear from the paper or fragpipe’s website where the standalone MSBooster tool is

Response: Thank you for bringing this to our attention. We added the link to MSBooster’s GitHub page in the “Code Availability” section of the paper, although the standalone version is not yet available to the public. We also have included some discussion of a standalone version in the Discussion section (line 472), as well as instructions in the software demo.

Manuscript changes: Second, we plan to extend MSBooster's flexibility via the availability of a standalone command line version to be incorporated into various pipelines outside of FragPipe. This would be applicable for users interested in de novo sequencing [78-80] or using MSBooster in conjunction with other PSM rescoring tools besides Percolator [75, 81].

4. I would highly recommend enabling output beyond the `pin` file. There are other means to filter PSMs beyond percolator (and new methods that folks will try in the future) and these would need a flatfile type of some sort that is more generic.

Response: Thanks for the suggestion. We recognize that the pin file format is widely used for its conciseness and informative nature. Other tools can easily adopt this format for input due to its simplicity and popularity. However, we plan to directly add calculated features to pepXML in the future. This has been added to the future directions in the Discussion section, line 475.

Manuscript changes: Also, as these interests arise, MSBooster can be adapted to work with PSM table formats besides Percolator pin files, such as pepXML files.

5. For the IM data, what input can be used? Such as, would DMS/FAIMS data be enabled instead of tims?

Response: MSBooster can still be utilized with FAIMS data by incorporating spectral and RT features, but because FAIMS CV values are not predicted, no IM features are calculated for FAIMS data.

6. "...this may simply reflect the moderate performance of MaxQuant in nonspecific searches..." The authors should include a comparison to MaxQuant rather than a reference to make the comparison "fair" here.

Response: Thank you for the suggestion. We have added specific numbers (159%) for MaxQuant and Prosit's performance on a prior dataset with 95 monoallelic cell lines (line 194), as well as performance of MaxQuant with/without Prosit on the Klaeger et al. dataset presented in the paper, on line 211. The Methods section has been updated with details of how MaxQuant and Prosit were run (line 582-595, line 607-619).

Manuscript changes: HLA peptide rescoring with DL features has previously been explored in [46]. Wilhelm et al. showed an average increase of 159% in peptide identification across 92 monoallelic cell lines [58] when using MaxQuant coupled with Prosit rescoring (Supplemental Fig 3a).

We then specifically compared performance between MSFragger/MSBooster and MaxQuant/Prosit on the Klaeger et al. data. MaxQuant initially reported 1569 peptides. After PSM rescoring with Prosit, it reported 10680 peptides, a 681% increase. This is compared to the 10138 peptides reported with MSBooster rescoring averaged across ten Percolator runs (Supplemental Fig 3b). To see if MaxQuant's performance could be

attributed to Prosit's predictions, we configured MSBooster to accept Prosit-predicted spectral libraries (Methods). Indeed, using Prosit in lieu of DIA-NN allowed MSBooster to achieve 10798 peptides on average, 118 more than MaxQuant with Prosit (Supplemental Fig 3b).

MaxQuant search and FDR control. MaxQuant v2.1.0.0 [9] was used to search the HLA immunopeptidome data [57]. Search tolerance was 20 ppm. For the MaxQuant only search, oxidation of methionine, n-terminal acetylation, pyroglutamation of glutamine and glutamic acid, and cysteinylation minus carbamidomethylation of cysteine were specified as variable modifications. Carbamidomethylation of cysteine was specified as a fixed modification. FDR at all levels was set to 0.01. For the MaxQuant search to be used for Prosit rescoring, because the only PTMs supported by the base non-TMT Prosit model are carbamidomethylation of cysteine and oxidation of methionine, the former is set as the only fixed modification, the latter as the only variable modification. All FDR levels were set to 1, as is required for Prosit rescoring.

MaxQuant analysis of Sarkizova et al.

Supplemental data 3 was downloaded from [46]. For each allele, the number of peptides with non-NA scores were counted and compared before and after Prosit rescoring. The average across all cell lines was calculated. Results are shown in Supplemental Fig 3a.

Prosit [46] was used for rescoring of HLA immunopeptidome data [57], both with MaxQuant and MSBooster. When running with MaxQuant output, the "rescoring" pipeline at <https://www.proteomicsdb.org/prosit/> was used. The msms.txt from the MaxQuant with FDR = 1 search was used as input, along with the individual RAW files. The size of the unique set of peptides with q-values less than 0.01 are shown in Supplemental Fig 3b. When used in combination with MSBooster, a command line version of MSBooster was used to extract peptides from the pin files, analogous to how it is done in FragPipe with DIA-NN. The peptides are formatted into an input file for the "spectral library" pipeline in <https://www.proteomicsdb.org/prosit/>. For both MaxQuant and MSBooster, the "Prosit_2020_intensity_hcd" and "Prosit_2019_irt" models are used for MS/MS spectral and RT predictions, respectively. The resultant "msp" file from Prosit is read by MSBooster, and spectra for peptides with PTMs besides carbamidomethylated cysteine and oxidized methionine are generated via m/z shifting, analogous to how it is done when used in conjunction with DIA-NN.

7. Did the DIA data have IM features? If not have the authors tried running these comparisons using IM-DIA data?

Response: Our DIA melanoma dataset did not include IM features. diaPASEF is not yet supported in FragPipe.

8. P7, lines 212-214. Can you add the raw numbers in addition to the percentages and note that 80% is a sum of the high/weak binders?

Response: These changes have been made.

9. "This indicates that Percolator gives little preference to higher-ranking PSMs in DIA data..." Is rank weight masked by other factors? For example, if hyperscore and rank are strongly related. Also, how is this affected by the inclusion of only rank 1 PSMs in the regression (p16, line 506)?

Response: The reviewer is indeed correct that hyperscore and the DL features influence Percolator's linear SVM weights. We have added "rank" into the correlation matrix in Supplemental Figure 2b and show that it is most highly correlated with hyperscore (absolute Spearman's R of 0.51). When removing hyperscore and the DL features from Percolator, the rank variable has a weight of ~-0.3, indicating a slight preference for higher ranked (rank 1) PSMs. We have removed discussion of the rank feature from the results, as this was convoluted by the other Percolator features that were correlated to it.

10. "Overall...IM feature scores...not sufficient to warrant their use in MSBooster." From Figure S5b, it appears that IM data and mass to separate decoys with the main group of decoys coming from 1+ precursors. A linear model accounting for both parameters in a single value might be enough to make the IM data more useful and seems worth trying.

Response: Thank you for the useful suggestion. Because Percolator is already a linear SVM, we tested adding raw 1/K0 and one-hot-encoded charge values (5 columns from 1 to 5, with values either being 0 or 1) to Percolator. Our results are presented in Supplemental Figure 11, and indeed they perform slightly better than our deep learning-based IM feature. We include this briefly in the Discussion section (line 460-462), with full description in Supplemental Notes 2

Manuscript changes: In addition, we compared our IM feature to using only raw 1/K0 and charge values for rescoring and found that the latter had better performance. We present the results in Supplemental Note 2 and Supplemental Fig 10-11.

11. Similarly, does the use of IM features change the weights for charge?

Response: Our hand-crafted IM feature "IM probability uniform prior" did not show any Spearman's R higher than 0.5 between itself and the one-hot-encoded charges. Raw 1/K0 values and charge 1 were correlated with R=0.68. When used together, 1/K0 weights change from negative (lower 1/K0 for targets) to positive values; the weight for the charge 1 feature changes from -0.2 to -1. The weights themselves are hard to interpret because the high correlation poses problems for the learned weights from the linear SVM Percolator, although the actual performance of target-decoy separation improves. Therefore, we abstain from discussion of the feature weights in the manuscript, as the collinearity problem makes interpretation near impossible.

12. What is the "grey zone"?

Response: The grey zone was referring to PSMs that change which side of the q-value threshold they fall in when they are rescored with MSBooster. It was an ambiguous phrase, and we have removed this phrase and replaced it with more understandable wording on line 441-442.

Manuscript changes: *We observed robust gains across applications, especially in analyses exhibiting a large search space.*

13. "...MSBooster can be used to process the MSFragger output for DDA data with multiple PSMs reported per spectrum...". This is potentially very exciting, but if it's claimed then there should be data to back it up. Please add a figure for this.

Response: We agree with the reviewer that this could be very exciting. This approach is primarily employed to identify co-fragmented peptides. However, as MSFragger currently does not support chimera spectra, we have decided to rephrase this statement to reflect a future direction for our research on line 465-470. We appreciate the reviewer's suggestion and will consider presenting results related to this aspect once the necessary support becomes available.

Manuscript changes: *For example, tools such as ionbot and CHIMERYS can report multiple PSMs per spectrum, potentially allowing consideration of more candidate sequences or co-fragmenting peptides [76, 77]. MSFragger could be optimized for identification of co-fragmenting precursors present in DDA data, and MSBooster could be adapted to rescore multiple peptides reported for chimeric spectra, in a similar fashion to rescoring multiple ranks in DIA data.*

14. Supplementary Figures. There seems to be an issue with the supplementary figures as they are at low resolution which makes interpretation nearly impossible for some (Figure S2, S6, S7).

Response: Thank you for noting this. We believe something went wrong with our file conversion. We have included all figures including supplemental in a Word document and checked that the resolution is acceptable.

Minor Concerns and Questions

- "Eluding" is used multiple times instead of "eluting". E.g., p3, line 54.

Response: Thank you for catching our spelling errors. These have been corrected in the introduction.

- P3, line 62. Please correct "data independent acquisition (DDA)".

Response: We have corrected this to "data dependent acquisition".

- There are a lot of parentheses in the introduction (sometimes two to three in one sentence: p4, lines 82-87) that suggest the sentence structure could be rearranged to make some sections more clear.

Response: We have changed the sentence structure to be more concise.

- P7, line 200: "...FDR were shared regardless of [whether] MSBooster was used..."

Response: We have added "whether" to the sentence

Reviewer #2 (Remarks to the Author):

This is a well written article that proposes a relatively straightforward idea and tests it thoroughly and convincingly. Certainly, the idea that deep learning models can be used to help improve statistical power to detect peptides from mass spectrometry data is not novel. And the approach proposed here -- augmenting the feature vector that goes into Percolator with a collection of scores from a pre-trained deep learning model -- is neither super novel nor profound. But it's also the most straightforward way to modify existing pipelines like MSFragger and, as the authors show here, it works quite well in practice.

The introduction is very nicely written, giving a brief but accurate synopsis of the field and of this work's contribution to it. I liked that the results section concisely covers a diversity of applications, from HLA peptide identification to DIA analysis to single-cell analysis and even timsTOF data. In each case, the analysis shows convincingly that using the deep learning features boosts power to detect peptides, to varying degrees. I have only a few relatively minor critiques.

Response: We thank the reviewer for the kind words. We agree that our approach is straightforward and applicable in many analyses. We provide a point-by-point response and highlight in the manuscript where we have addressed your critiques.

It's not accurate to say that existing post-processing algorithms "do not incorporate prior knowledge regarding peptide separation coordinates" (line 95). Until recently, Percolator included a specific set of features to model RT (see Moruz JPR 2010). This same paragraph later addresses this point, and even cites the Moruz paper, though it implies incorrectly that only PeptideProphet has incorporated such estimates into its models.

Response: We thank the reviewer for pointing out this discrepancy. We have edited that paragraph on line 101-103 to properly credit Percolator for integrating RT information into modeling.

Manuscript changes: The difference between the experimental and predicted retention times is also known to provide additional discriminating power [31-34]; RT differences have previously been incorporated into Percolator and PeptideProphet modeling [34, 35].

In the description of MSFragger searching, I am guessing that the software includes predefined sets of parameters, referred to here as "workflows." This point needs to be clarified in the text, because this will not be obvious to people who have not used your software before.

Response: The reviewer's description is accurate. We agree that this is not immediately clear in the manuscript, so we have added a brief description at line 149-152 of what a

workflow is, along with values of the essential parameters of each workflow (line 558-574).

Manuscript changes: Within FragPipe, MSBooster is positioned between MSFragger and Percolator (Fig 1a) and is enabled by default in most FragPipe analysis workflows (see Methods section for details), where a FragPipe workflow is the order in which software are to be executed, along with optimized parameters for each tool.

The workflows used for each dataset are as follows: HLA immunopeptidome [57] (nonspecific-HLA-C57 workflow); melanoma neoantigen [65] (nonspecific-HLA with carbamidomethylated cysteine added as a variable modification); melanoma DIA data [66] with MSFragger-DIA (DIA_SpecLib_Quant) and with DIA-Umpire (DIA_DIA-Umpire_SpecLib_Quant); HeLa timsTOF [70], single cell proteomics with nanoPOTS [67] or DISCO [45], and secretome [69] (Default). All workflows included oxidation of methionine and N-terminal acetylation as variable modifications. The workflows besides Default also included pyro-glutamation from glutamine and glutamic acid. The HLA workflow had carbamidomethylated cysteine as a fixed modification with the mass difference between cysteinylation and carbamidomethylation (61.98 Da) as a variable modification; the neoantigen workflow included both carbamidomethylation and cysteinylation as variable modifications. A maximum of 3 variable modifications was allowed. Peptide length was set to 7-25 for nonspecific workflows and 7-50 for all others. All workflows used 20 ppm for precursor and fragment error tolerance, with mass calibration and parameter optimization enabled. MSBooster, Percolator, ProteinProphet, and Philosopher were enabled. The HLA workflow was revised to add “—mods M:15.9949” to the Philosopher filter to perform group-specific FDR estimation [87] using the following three categories: unmodified peptides, peptides with oxidized M only, and peptides with any other modification.

The paragraph beginning on line 218 talks about how MSBooster handles PTMs that are not supported by DIA-NN. This is mildly interesting, but I don't think the analysis (Supp Fig 4) directly shows what you claim, namely, that "MSBooster allows re-scoring of any PTM-containing peptide, even if a small penalty is applied in the case of certain PTMs (not yet supported by DIA-NN prediction module)." To me, what would be interesting is information about the numbers of such modifications that are detected with and without the DL features. The analysis in this paragraph really only shows that such PTMs receive a small penalty in score.

Response: We appreciate the reviewer's input, prompting us to delve deeper into the analysis of modified peptides. We have conducted a thorough examination to assess the impact of uncommon modifications on prediction accuracy and identification results. Intriguingly, we observed that Pyro-Glu from Q was significantly affected after rescoring (Supplemental Fig 4). In light of this, we have revised our statements on line 258-268 to acknowledge that MSBooster's handling of PTMs is advantageous for those whose properties align closely with DIA-NN's predictions; however, it may be suboptimal for certain PTMs. We recognize this limitation and intend to enhance our handling of new PTMs in future iterations of the tool.

Manuscript changes: *The RT shift in pyro-Glu peptides is expected and has been previously recapitulated [64]. While most PTMs showed an increase in peptides containing those modifications (Supplemental Fig 4c), the exception is pyro-glutamation from Q, where reported peptides dropped from 105 to 25. This decrease is mainly driven by the increased RT difference, as no unique peptides are reported after only rescoring with the RT feature, while rescoring with only spectral similarity results in only 32 lost peptides and 9 gained peptides. This is in stark contrast to the 82 lost and 2 gained peptides after using the MSBooster default of spectral and RT feature rescoring. Overall, our analysis shows that while MSBooster will not exclude any PTM-containing peptide, those PTMs not yet supported by the DL prediction module that drastically affect the peptides' physicochemical properties will be heavily penalized.*

The obvious question posed by Figure 3A-D is why 3B/D are included at all. The conclusion (which was already apparent from the MSFragger-DIA paper) is that DIA-Umpire does not work as well as MSFragger-DIA. Given this, I would recommend dropping 3B/D or, at best, moving them to a supplementary figure.

Response: Thanks for the suggestion. We have moved the DIA-Umpire results to the supplemental figures.

line 504: Please tell us what "preset threshold" is used in this work.

Response: A threshold of $10e-3.5$ was chosen and has been added to the Methods on line 630-633.

Manuscript changes: *To train the regression models, a subset of PSMs (5000, by default) with expectation values below a preset threshold ($10e-3.5$) is used.*

line 525: Also tell us the weight of the uniform prior.

Response: A description of how the uniform prior weight is calculated, and more generally how the entire feature is calculated, is now in the Methods on line 669-675.

Manuscript changes: *The uniform prior is decided by sorting the RT bins in ascending order by the number of PSMs contained in each. A bin is chosen based on a preset percentile (10^{th} percentile, by default). The number of PSMs U at the RT bin at this specific percentile is chosen. The uniform prior probability P_U is distributed equally across the predicted RT range. If a PSM with empirical probability P_E is placed in a bin that contain E PSMs, the value of the "RT probability uniform prior" feature is*

$$\frac{P_U * U}{U + E} + \frac{P_E * E}{U + E}$$

In the supplemental note, please be more precise about the RT loess term calculation: "some subset of PSMs with lowest expectation values" is not specific enough. Is there a rank threshold involved? Also, please specify how the calibration of experimental to predicted RT is done exactly. For the "RT probability unif prior," specify exactly how the Gaussian smoothing is done, and how the kernel mean is selected. I don't actually understand this sentence: "A PSM can be rescored by matching its experimental RT to the appropriate bin and calculating the probability from the empirical distribution." Please expand this description.

Response: We have added more specific details about the RT feature calculation steps, starting at line 628. Because KDE generates a Gaussian curve for each point, the mean of each curve added is the value of the point the curve is centered on. These details of the RT feature calculation are now all presented in the Methods, so all details can be found in one central location rather than split between Methods and Supplemental Notes.

Minor:

In Figure 1, it seems strange to me that there is an arrow from "PSM feature tables" to Percolator. Don't the "Extended PSM feature tables" contain all the features?

Response: The reviewer is correct. The extended PSM feature tables include everything reported by MSFragger (PSM feature tables) and the new deep learning features from MSBooster. There is an arrow there to illustrate Fig 1a's original pipeline, where PSMs are passed directly from MSFragger to Percolator. In Fig 1b with MSBooster, the extended tables are now passed to Percolator.

line 185: I assume that these results are at a 1% peptide-level FDR, but I think this should be stated explicitly.

Response: This was at 1% FDR. We have now added clarification of this the first time we mention peptide/protein numbers, as long as a clarification in the "MSFragger search and FDR control" subsection of the Methods section.

line 200-201: Missing word ("whether")

Response: We have added "whether" to the sentence.

Line 334: I think you should define "1/K0" the first time it is used.

Response: Inverse ion mobility (1/K0) has been added for clarity.

line 431: searches -> searched

Response: This has been fixed.

Reviewer #3 (Remarks to the Author):

The manuscript, "MSBooster: Improving Peptide Identification Rates using Deep Learning-Based Features" by Yang et al., reported the advantage of combining the deep learning (DL) module of "DIA-NN" that was originally reported by the two authors of this paper (Demichev, V., et al., Nature Methods, 2020) and "MSFragger" that was reported by the last author of this paper (Kong, A.T., et al., Nature Methods, 2017). The authors used their knowledge of coding and MS-search to construct DL-assisted "MSBooster" and validate its utility by using existing repository datasets.

Specifically, they used monoallelic immunopeptidomics, DIA proteomics, single-cell proteomics, and timsTOF datasets, showing that RT and spectral data improved peptide identification efficiency. The codes that the authors have already established in the past (DIA-NN and MSFragger) are both excellent, and the challenge of combining the two to further improve identification efficiency is commendable. On the other hand, in light of the academic impact of the journal, it can not be denied that the content of this manuscript lacks the novel findings obtained by combining the MSFragger with DIA-NN. Given the current increasing reports on the construction of similar algorithms and the launch of commercially-available DL-assisted software, additional validation is still necessary to clarify the advantage of MSBooster.

Response: We thank the reviewer for the commendation. The reviewer is correct that there are many tools available for PSM rescoring with deep learning predictions. We would like to emphasize that one of the novelties of MSBooster in FragPipe is the ease of use, which we have now highlight throughout the text. Many users have already incorporated MSBooster into their FragPipe analyses due to its ease of use. This is in contrast with tools that require external prediction of spectral libraries, such as MaxQuant with Prosit. We have streamlined this process, allowing easy access for researchers with less computational experience. Commercial tools such as those from MSAID can be costly to academics, and they require uploading data to the cloud, potentially bringing issues of data security. Therefore, apart from the many scenarios in which MSBooster fits in seamlessly, we present this fully integrated solution as a major innovation presented in our paper.

Major Comments

1. From Figures 2-4, it appears that the improved efficiency of peptide identification is mainly useful for immunopeptidomics (up to 31.4% increase by Spectra+RT similarity features, other datasets limited from a few to 16.6% increase at most). However, validation search by monoallelic sample (only with A*02:01 allotype) seems too simplistic to demonstrate the efficacy of MSBooster.

Immunopeptidomics is highly desired in clinical immunology in which datasets from human specimens are important. This means necessity of immunopeptidomics under full allelic conditions, including up to six HLA motifs. Therefore, it's better to show the results of validation using previously reported actual clinical datasets (e.g., Melanoma sample by Bassani-Sternberg, M. et al. Nat. Commun. 7, 13404 (2016), TNBC sample by Ternette, N. et al. Proteomics 18, 1700465 (2018), CRC sample by Minegishi, Y. et al. Commun. Biol. 18;5(1):831

(2022), etc.) to determine the degree to how much the efficiency of peptide identification can be improved by MSBooster.

Response: We thank the reviewer for advocating for a more difficult and relevant clinical example. We have tested and added rescoring results for the Mel15 sample from Bassani-Sternberg et al, presenting the results in a whole new section entitled “Neoantigen Discovery” starting on line 270. We note that while further wet lab experiments are necessary to validate our findings, MSBooster does improve our ability to find previously reported and novel neoantigens.

Since the DL-assisted search software (Proteome Discoverer 3.0, etc.) as well as the immunopeptide-specific de novo sequencing software (DeeNovo by PEAKs, etc.) have commercially launched, it is necessary to demonstrate and emphasize the scientific advantage of MSBooster. For that, if neoantigens that could not be retrieved by the past search methods could be identified from the reposit datasets, I believe that this manuscript is worthy of publication.

Due to the current upgrowth of reports on the construction of algorithms that claim to improve efficiency of peptide identification, additional clear examples except for immunopeptidomics should be crucial to show the feasibility of what new findings can be retrieved by MSBooster.

Response: Full integration of MSBooster into FragPipe and its interplay with other tools in FragPipe is a scientific advantage of our approach. In addition to boosting HLA identifications, MSBooster gives users full access to view predictions from DIA-NN via the many graphics generated with the help of MSBooster. For example, spectral and RT similarity scores are now added to our results files from Philosopher; PDV visualization enables users to compare experimental and predicted spectra; and MSBooster produces RT calibration curves to show how well predicted and experimental RT values align (Supplemental Fig 13).

We could not compare our findings to PD or PEAKS due the need for a commercial license. However, we have compared our findings with those from the original Bassani-Sternberg study and the rescoring results from DeepRescore on line 286. We report a neoantigen unique to our analysis with similar MSBooster scores as other repeatedly reported neoantigens, as well as predicted binding affinity.

*Manuscript changes: We also compared our identified neoantigens to those reported in prior analyses of the same data [49, 65] (Fig 3c). Consistent with prior work, we only considered length 8-12 peptides here. Both we and DeepRescore [49] rejected two peptides reported in the original Bassani-Sternberg study [65] – ASWVVPIDIK, which MSFragger did not report, and GRTGAGKSFL (MS2 similarity: 0.81, delta RT: 8.77), which did not pass 1% FDR. Similarly, DeepRescore suggested two peptides that MSBooster did not – DVFPEGTRVGL, which MSFragger did not report, and RLFLGLAIK (MS2 similarity: 0.74, delta RT: 4.12), which did not pass 1% FDR (Supplemental Data 2). MSBooster reported one unique peptide in the allowed length range, SLSSALRPSTSR. The best spectral similarity across all PSMs of this peptide was 0.9872, and the lowest delta RT was 0.7127 iRT (Fig 3d). Its predicted binding affinity to one of Mel15’s alleles A*03:01 was 1468.73 nM, designating it as a weak binder (Fig 3e).*

2. As authors mentioned by themselves, since the ion mobility works on the MS1 precursor ion separation, the DL algorithm in MS2 fragment ions does little in improved identification. So, the effect of using DL-assisted search by ion mobility does not seem to be a matter mentioning in result section at least in this manuscript.

The result section of timsTOF PASEF data with ion mobility can be briefly mentioned in discussion section (Tried to introduce IM factor into DL features but didn't work because IM works on MS1 separation, etc.) and can replace the Figure 5 into Supplementary information.

Response: We thank the reviewer for the suggestion to declutter the main text. We believe that this section is important to present briefly in the main text because it also showcases how MSBooster performs on tryptic DDA bulk-cell data, which is one of the simplest analyses to perform in FragPipe. MSFragger performs quite well even without deep learning features, and we wanted to showcase that in this example. We also believe this to be worth presenting in the main results section as it is contradictory to prior proprietary work that suggests DL IM features can substantially increase peptide IDs (<https://www.bruker.com/ru/news-and-events/news/2022/bruker-releases-ccs-enabled-timscore.html>). We do move much of the IM results to Supplemental Notes 2. We have clarified that the DL predictions are also for 1/K0 of the MS1 precursors (line 393-394).

Manuscript changes: *As such, DL models have been extended to predict ion mobility or related collisional cross section values [38, 43] for peptide ions.*

3. The obtained MS2 spectra are different by distinct collision modes. The detectors of MS2 are also different from mass spectrometer to the other. The ion mobility-assisted MS can be conducted by not only timsTOF but also by the FAIMS-interface. As such, the data acquisition conditions on the MS side can be a rate limiting factor for peptide search by each algorithm. Therefore, it is informative to clearly mention the details of MS parameters; collision mode (HCD/CID/EThcD, etc.), MS2 detector (TOF/Orbitrap/Iontrap, etc.), with or without IM options (timsTOF or FAIMS interface, etc.) in each dataset used in the validation study. Then comparative study of which MS conditions can be the most beneficially MSBooster-search can clearly demonstrate the further superiority of this algorithm.

Response: We have added details of collision mode, mass spectrometer/mass analyzer, and ion mobility options when we present each dataset. We agree that a comparative study of how MS conditions affect MSBooster performance would be beneficial to know what to expect from gains in IDs. This could be extended to a simple multiple regression model to predict percent gains based on experimental metadata and search settings. This would be most possible with access to data from the same lab, where the researchers keep most constants the same, but change a few parameters at a time such as mass spectrometer or LC gradients. We include a discussion of this as a future direction on line 446-451, but with the current data we have analyzed, we do not believe we can draw any concrete conclusions or trends, as each subsection of the results deals with a different kind of data analysis; it is hard to compare MS conditions between the HLA and timsTOF data because of the confounding factors of the database search settings and types of peptides analyzed.

Manuscript changes: *Several factors may contribute to MSBooster performance, including MS2 spectral quality, deviation between the experimental data acquisition*

parameters and those of the training data for the prediction models, and the number of PSMs available for rescoring. We present general guidelines for what level of gains are expected from multiple popular applications of MSBooster, but an in-depth analysis of how each of these characteristics of the data is outside of the scope of this study.

Minor Comments

1. In main text, line 294-296, “Surprisingly, when looking at the different cell numbers separately, we did not notice a clear relationship between the cell number and MSBooster performance (Fig 4c-d)”. What does this “a clear relationship” mean? Didn't this simply mean the search by MSBooster for nanoPOTS dataset did not yield increased results corresponding to the increased cell counts? Doesn't this simply mean the MSBooster does not work reliably with datasets that contain too little information, such as single cell proteomic by nanoPOTS data?

Response: We have changed “clear relationship” to “monotonic relationship”, as one might hypothesize with increasing spectral similarity distributions with increasing cell number. This lack of monotonicity may be a result of several factors affecting the data quality.

Or is it because the cellular proteome, which is the primary purpose of single cell analysis, is incompatible with the system of MSBooster's rescoring because of the distinct major protein expressions in each cell types included in the target sample?

Response: It is possible that random differences between which proteins and what number of proteins each cell is expressing at a particular moment may differ between single cells, and this stochasticity is more apparent when a small number of cells are present. It is also possible that random variation between quality of the runs and batch effects contribute to the seemingly random gains across different numbers of cells.

Wasn't that why the results by MSBooster didn't correspond to the number of cell counts? Or, the authors meant the data acquisition by nanoPOTS is qualitatively poor (or not applicable for MSBooster?) compared to the DISCO method, that was why no clear relationship between the cell number and MSBooster performance?

Response: The DISCO data had greater spectral similarity compared to the nanoPOTS data. However, this is not an indication that nanoPOTS quality is low, or that MSBooster is not applicable. MSBooster relies on the quality of spectral and RT predictions, and single cell spectra have been shown to differ from bulk cell spectra (see Discussion on line 494-499). This difference in spectra is more apparent in the nanoPOTS data, but could be ameliorated with models specifically for single cell or models fine tuned to the MS conditions of the nanoPOTS data. This is discussed in the future directions, as new models besides DIA-NN may take into account conditions such as mass spectrometer and collision energy, which are not currently considered by DIA-NN prediction.

Manuscript changes: For example, while single and bulk cell spectra appear similar on a timsTOF Pro instrument [84], they appear different enough on an Orbitrap instrument that one may consider a model tuned for single cells [68]. Different fragmentation

mechanisms, mass spectrometers, and collision energy settings also impact MS/MS spectra. These factors are not currently considered by DIA-NN peptide prediction, but they can have noticeable effects on spectra.

As I asked in the major comment #2, it is important to mention the compatibility of MSBooster with the conditions/parameters of data acquisition by MS-side. Please clarify and explain all these points in the main text.

Response: We have added details of conditions of data acquisition to this section on lines 339-340 and line 362-364.

Manuscript changes: We tested MSBooster on single cell data from the nanoPOTS platform [67] generated using an Orbitrap Fusion Lumos Tribrid instrument with HCD fragmentation.

we tested another dataset produced on a Q Exactive MS with Orbitrap mass analyzer with a different sample processing protocol (DISCO) [45] and HCD fragmentation

2. In main text, line 312, “10-15%” is an overstatement. In Figure 4g-h, maximum increase was described as 12.4% (Fig. 4g, spectraRT). This is only 12% when rounded off. Please correct the description accurately according to the data, or avoid overstatement.

Response: We have corrected the numbers to reflect the reported numbers in the figures (9.2-12.4%).

3. In Figures 2a, 3a-d, 4c-d, 4g-h and 5a-b, it seems that the statistical evaluation can be feasible for the rate of increase in identification by analysis using MSBooster. Please consider to add the statistical analyses to show the advantage of MSBooster clearly.

Response: We have performed the appropriate variants of t-tests for each dataset to show the improvement of sequentially adding DL features. These results are presented in Supplemental Data 1. As other papers that introduce DL PSM rescoring do not include significance tests in their figures, we leave them out of figures.

4. The resolution of Supplementary figure 2 is too poor to read the embedded-text. Please use the higher resolution, or add readable text in a larger font size.

Response: Thank you for noting this. We believe something went wrong with our file conversion. We have included all figures including supplemental in a Word document and checked that the resolution is acceptable.

REVIEWERS' COMMENTS

Reviewer #1 (Remarks to the Author):

The authors present a revised manuscript for MSBooster. I believe this manuscript is publishable now, but would benefit from inclusion of an example before and after pin file in the GitHub page. This would help users determine what their input data should look like and what they should expect as output.

Reviewer #3 (Remarks to the Author):

New impressive datasets, such as the "Neoantigen discovery" paragraph based on the reanalysis of Mel15 samples, have been added, which reinforce assertion of usefulness.

Other explanations have been also added to better understand the strengths and characteristics of the technique developed by the authors, and it is appreciated that the content sufficiently conveys the value of using this method for many readers.

I hope that the authors will continue to develop and disseminate less expensive, easy-to-use, and high-performance proteome analysis technologies in the future.

Reviewer #1 (Remarks to the Author):

The authors present a revised manuscript for MSBooster. I believe this manuscript is publishable now, but would benefit from inclusion of an example before and after pin file in the GitHub page. This would help users determine what their input data should look like and what they should expect as output.

Response: We thank the reviewer for accessing our GitHub documentation. We have added the expected input/output formats. We will continue to update the MSBooster documentation as the software matures.

Reviewer #3 (Remarks to the Author):

New impressive datasets, such as the "Neoantigen discovery" paragraph based on the reanalysis of Mel15 samples, have been added, which reinforce assertion of usefulness.

Other explanations have been also added to better understand the strengths and characteristics of the technique developed by the authors, and it is appreciated that the content sufficiently conveys the value of using this method for many readers.

I hope that the authors will continue to develop and disseminate less expensive, easy-to-use, and high-performance proteome analysis technologies in the future.

Response: We thank the reviewer for the suggestions to showcase our ability to identify neoantigens and to note MSBooster's generalizability to various instrument settings. We are excited to continue supporting and extending the capabilities of MSBooster.